# Excitatory-inhibitory homeostasis and bifurcation control in the Wilson-Cowan model of cortical dynamics

**Francisco Páscoa dos Santos** [1,2]*, **Paul F. M. J. Verschure** [3]

**1** Eodyne Systems SL, Barcelona, Spain, **2** Department of Information and Communication Technologies, Universitat Pompeu Fabra (UPF), Barcelona, Spain, **3** Donders Institute for Brain, Cognition and Behavior, Radboud University, Nijmegen, The Netherlands

* f.pascoadossantos@gmail.com

**Data Availability Statement:** All the code necessary to run the models can be consulted in:

## Abstract

Although the primary function of excitatory-inhibitory (E-I) homeostasis is the maintenance of mean firing rates, the conjugation of multiple homeostatic mechanisms is thought to be pivotal to ensuring edge-of-bifurcation dynamics in cortical circuits. However, computational studies on E-I homeostasis have focused solely on the plasticity of inhibition, neglecting the impact of different modes of E-I homeostasis on cortical dynamics. Therefore, we investigate how the diverse mechanisms of E-I homeostasis employed by cortical networks shape oscillations and edge-of-bifurcation dynamics. Using the Wilson-Cowan model, we explore how distinct modes of E-I homeostasis maintain stable firing rates in models with varying levels of input and how it affects circuit dynamics. Our results confirm that E-I homeostasis can be leveraged to control edge-of-bifurcation dynamics and that some modes of homeostasis maintain mean firing rates under higher levels of input by modulating the distance to the bifurcation. Additionally, relying on multiple modes of homeostasis ensures stable activity while keeping oscillation frequencies within a physiological range. Our findings tie relevant features of cortical networks, such as E-I balance, the generation of gamma oscillations, and edge-of-bifurcation dynamics, under the framework of firing-rate homeostasis, providing a mechanistic explanation for the heterogeneity in the distance to the bifurcation found across cortical areas. In addition, we reveal the functional benefits of relying upon different homeostatic mechanisms, providing a robust method to regulate network dynamics with minimal perturbation to the generation of gamma rhythms and explaining the correlation between inhibition and gamma frequencies found in cortical networks.

## Author summary

We study how excitatory-inhibitory (E-I) homeostasis controls edge-of-bifurcation dynamics in cortical networks and how it impacts the generation of gamma oscillations. Importantly, while previous studies have limited E-I homeostasis to the plasticity of inhibition, we explore the wide range of mechanisms employed by cortical networks and, more importantly, how they interact. Here, we derive the mathematical solution for the

https://gitlab.com/francpsantos/wc_node_homeostasis.

**Funding:** FPS was funded by euSNN project (Erasmus+ MSCA- ITN ETN H2020dID 860563). PFMJV was funded by the European Commission through AISN (Horizon Europe, 101057655), EBRAINS-HEALTH (Horizon Europe, 101058516), PHRASE (European Innovation Council, 101058240), ReHyb (Horizon2020 871767) and euSNN (Erasmus+ MSCA- ITN ETN H2020dID 860563). The funders played no role in the study design, data collection and analysis, decision to publish, or preparation of the manuscript.

**Competing interests:** I have read the journal's policy and the authors of this manuscript have the following competing interests: FPS was employed by Eodyne Systems SL. PFMJV is the founder and shareholder of Eodyne Systems S.L., which brings scientifically validated neurorehabilitation and education technologies to society.

Wilson-Cowan model under distinct modes of homeostasis and study how they shape model dynamics and the generation of gamma oscillations. That said, we demonstrate that E-I homeostasis, particularly of excitation and intrinsic excitability, modulates model dynamics relative to the bifurcation between damped and sustained oscillations in a manner previously unaccounted for, providing a mechanism for the implementation of heterogeneous distances to the bifurcation across cortical areas. Furthermore, our results stress the functional benefits of relying on multiple modes of homeostasis, allowing for the control of firing rates and circuit dynamics while ensuring that gamma oscillations remain within a physiological range and explaining the relationship between inhibition and gamma frequencies found in empirical data. With these results, we unify E-I balance, edge-of-bifurcation dynamics, and gamma oscillations under the lens of firing-rate homeostasis.

## Introduction

The human cortex comprises repeated motifs of excitatory-inhibitory (E-I) interactions [1–3]. For this reason, cortical excitatory pyramidal (PY) neurons are constantly bombarded by excitatory and inhibitory inputs, mostly from neurons in local circuits [4–8]. More importantly, there is extensive evidence indicating that the excitatory and inhibitory inputs to single neurons are, by and large, tightly balanced [9–12] and that ensuring balanced E-I interactions is essential for the operations that cortical networks perform [13–16]. In addition, there is strong evidence suggesting that cortical networks homeostatically regulate E-I balance, ensuring the stability of firing rates in PY neurons [17, 18]. The precise mechanisms of E-I homeostasis can take many forms, from synaptic scaling of excitatory [19, 20] and inhibitory synapses [21, 22] to the plasticity of the intrinsic excitability of PY neurons [23–25]. In addition, the mechanisms that maintain the stability of PY firing rates have been found to contribute to the self-regulation of cortical networks toward criticality [18, 26]. This is particularly relevant since E-I balance is linked to edge-of-bifurcation dynamics [26–29], which are thought to be a foundational mechanism underlying the emergence of collective dynamics, such as metastability, in the cortex [27, 30–34]. Furthermore, recent studies suggest that distinct cortical areas might be poised in different dynamical regimes relative to the bifurcation [35], reflecting their position in the cortical hierarchy [7]. Importantly, this heterogeneity in the distance to the bifurcation of cortical areas has been suggested to contribute to the emergence of metastable large-scale dynamics [32], suggesting a functional relevance beyond the local circuits. However, it is not clear how local cortical networks self-regulate not only toward the edge of bifurcation, but also how this set-point is determined across cortical areas to ensure heterogeneity in their dynamical regime relative to the bifurcation. That said, understanding how cortical networks regulate edge-of-bifurcation dynamics through E-I homeostasis might be essential to understanding large-scale cortical dynamics.

In this context, several studies have endeavored to study E-I balance in large-scale brain models, demonstrating its role in supporting criticality [36] and metastability [37], and in shaping functional resting-state networks [37–39]. Furthermore, our results suggest that E-I balance is not only relevant for the emergence of spatiotemporal dynamics across timescales [40], but that E-I homeostasis can contribute to the recovery of functional networks following perturbations to the structural connectome [41]. There are, however, some caveats common to all these approaches. The first is that they commonly rely only on the regulation of feedback inhibition [36–39], even though it is known that E-I homeostasis is not limited to inhibitory

synapses alone [17, 18, 42]. More specifically, it can also operate through the modulation of excitatory synapses [19] or intrinsic excitability [23]. Furthermore, it has been speculated that the concurrent action of multiple homeostatic mechanisms is relevant to avoiding drastic changes in neuronal dynamics [43] and to maintaining network dynamics close to criticality [18, 42]. Importantly, our results suggest that homeostasis of inhibition alone is not sufficient for recovering network dynamics after lesions to the connectome [41]. However, to our knowledge, other modes of E-I homeostasis have not yet been validated in large-scale models. Furthermore, another gap in the literature is the lack of detailed exploration of the impacts of homeostasis on model dynamics. For example, manually regulating the synaptic couplings of the Wilson-Cowan (WC) model, a common neural-mass model relying on E-I interactions [44, 45], affects the frequency of oscillations and bifurcation dynamics of the model [46], suggesting that processes that modulate the same parameters, such as E-I homeostasis, might have similar effects. However, this possibility remains unexplored in the literature.

Given the foundational relationship between the dynamics of individual cortical areas and the emergent behavior of large-scale cortical networks, we aim to elucidate the impact of E-I homeostasis on the dynamics and bifurcation portrait of cortical areas. Therefore, in this paper, we study the behavior of the WC model under different forms of homeostasis by deriving analytically the fixed points of the different modes of E-I homeostasis and studying their impact on node dynamics using bifurcation theory. In addition, we devise a method to compute the model's fixed points when multiple modes of E-I homeostasis are implemented in conjunction, allowing for the study of the WC model under multiple modes of homeostasis and how their concerted action shapes oscillatory dynamics. Furthermore, we analyze the effects of each mode of homeostasis on node frequency, suggesting that the cooperation of multiple modes of homeostasis allows for minimal impact on node parameters, maintaining the oscillation frequency within the gamma range. More importantly, our results reveal how edge-of-bifurcation dynamics can be controlled in cortical networks through the homeostatic regulation of mean firing rate. In addition, some modes of homeostasis can modulate the distance to the bifurcation, depending on the magnitude of external inputs, while maintaining the same mean firing rate. Therefore, we advance E-I homeostasis as a mechanism not only for the maintenance of stable activity in cortical neurons, but also to ensure edge-of-bifurcation dynamics, as previously suggested [18, 26], offering an explanation as to why different cortical areas might be poised in different regimes relative to criticality [32, 35]. In summary, the work presented here not only provides a mathematical basis for implementing multiple modes of E-I homeostasis in the WC model, but also elucidates how cortical networks can maintain dynamics at the edge of bifurcation through E-I homeostasis and how the combination of multiple processes of E-I homeostasis can ensure the maintenance of stable firing rates with minimal impact on the intrinsic rhythms of cortical networks.

## Methods

### Wilson-Cowan model

The WC model represents the dynamics of reciprocal interactions between populations of excitatory and inhibitory neurons in the cortex [44, 45]. Given the relevance of excitatory synaptic scaling as a mechanism of E-I homeostasis [17, 19, 21, 22, 47, 48], we propose an updated form of the canonical WC model by adding a term $G^E$ to control the gain of all excitatory inputs (i.e. recurrent excitation and excitatory external inputs). Therefore, we consider the following model equations, describing the firing rates of coupled excitatory ($r^E(t)$) and inhibitory

$(r^I(t))$ neural masses:

$$\tau^E \frac{r^E(t)}{dt} = -r^E(t) + F^E(G^E c^{EE} r^E(t) - c^{EI} r^I(t) + G^E I^{ext}(t))$$

$$\tau^I \frac{r^I(t)}{dt} = -r^I(t) + F^I(c^{IE} r^E(t))$$

(1)

where $c^{EE}$ represents the recurrent coupling in excitatory populations, $c^{IE}$ represents the coupling from excitatory to inhibitory populations and $c^{EI}$ represents the coupling from inhibitory to excitatory populations. $I^{ext}$ represents an external excitatory input to the excitatory population, used here to represent the input received from other cortical areas, which is by and large excitatory [49–51]. The sigmoid activation (or input-output) functions $F^E(x)$ and $F^I(x)$ can be defined as:

$$F^{E/I}(x) = \frac{1}{1 + e^{-\frac{x - \mu^{E/I}}{\sigma^{E/I}}}}$$

(2)

where $\mu$ and $\sigma$ represent the firing threshold and sensitivity of the neural masses, respectively. Here, we consider that the activation functions of excitatory and inhibitory populations can be dissimilar and, thus, we define $F^E(x)$ with parameters $\mu^E$ and $\sigma^E$ for the excitatory neural mass and $F^I(x)$ with parameters $\mu^I$ and $\sigma^I$ for its inhibitory counterpart. This modification allows for the implementation of plasticity of intrinsic excitability of the excitatory population only [23, 47]. The default model parameters follow the implementation of [39–41] (Table 1). Unless stated otherwise, all parameters have arbitrary units. The time constants $\tau^E$ and $\tau^I$ were tuned so that our default model generates intrinsic oscillations at $\sim 40$Hz near the bifurcation. This was done to approximate the resonance of cortical networks at gamma rhythms through the interaction between PY cells and fast-spiking inhibitory interneurons [52–55], in line with previous modeling studies [40, 41, 56, 57]. On another note, it is relevant to point out that the original definition by Wilson and Cowan [44] also accounts for self-inhibition of the inhibitory population. However, to ensure that the model displays a Hopf-Bifurcation behavior between damped and sustained oscillations, conforming with the edge-of-bifurcation hypothesis [26–29], it must be guaranteed that the inhibitory-to-inhibitory coupling is close to 0 or, at least, considerably smaller than the rest of the couplings in the model [45]. Here, similarly to previous approaches [39–41], we set that parameter to 0. Nonetheless, we explore the effects of E-I homeostasis in models with different levels of self-inhibition ($c^{II}$) (S1 Appendix and S1 Fig). In addition, we also study the behavior of models under different levels of input to the inhibitory population (S1 Appendix and S1 Fig). Importantly, while we set both these parameters to 0 in

**Table 1. Default parameters of the Wilson-Cowan model.**

| Model Parameters | |
|---|---|
| $c^{EE}$ | 3.5 |
| $c^{IE}$ | 3.75 |
| $c^{EI}$ | 2.5 |
| $G^E$ | 1 |
| $\mu^E/\mu^I$ | 1 |
| $\sigma^E/\sigma^I$ | 0.25 |
| $\tau^E$ | 2.5 ms |
| $\tau^I$ | 5.0 ms |

the main text, the Hopf-bifurcation can still exist in circuits with self-inhibition and input to the inhibitory population, provided that both these values are kept within certain bounds. For a more detailed explanation, refer to S1 Appendix and S1 Fig. All the code necessary to run the models can be consulted in https://gitlab.com/francpsantos/wc_node_homeostasis.

## Derivation of nullclines and fixed points

The fundamental function of E-I homeostasis is to ensure the stability of firing rates in PY neurons by adapting their synapses or excitability [17]. In dynamical models, this translates into adjusting the parameters until the system reaches a stable state corresponding to the target activity levels. Therefore, we start by deriving the expression for the fixed points of the WC model. The first step is to obtain the expression for the system nullclines, which describe all the values of $(r^E, r^I)$ for which either $\frac{dr^E}{dt}$ or $\frac{dr^I}{dt}$ are equal to zero. The intersection of the system nullclines defines the system's fixed points, where both derivatives equal 0. The expression for the $r^E$ nullcline, obtained from Eqs 1 and 2, is the following:

$$r^I = \frac{\sigma^E}{c^{EI}} log\left(\frac{1 - r^E}{r^E}\right) + \frac{G^E c^{EE} r^E + G^E I^{ext} - \mu^E}{c^{EI}} \tag{3}$$

Similarly, we can derive the nullcline of $r^I$ by equating $\frac{dr^I}{dt}$ to 0, yielding:

$$r^I = \frac{1}{1 + exp(-(c^{IE} r^E - \mu^I)/\sigma^I)} \tag{4}$$

Here, $log$ denotes the natural logarithm and $exp$ the exponential function.

For more details on the derivation of the system's nullclines, refer to S2 Appendix. In Fig 1A, we plot the nullclines for three examples of the WC model under different levels of external input $I^{ext}$.

Considering the $r^E$ and $r^I$ nullclines, the fixed points of the system can be derived by computing the intersection of both lines, yielding the values of $(r^E, r^I)$ for which both $\frac{dr^E}{dt} = 0$ and $\frac{dr^I}{dt} = 0$. Therefore, the fixed point $r^E$ can be obtained by solving the following equality:

$$\frac{\sigma^E}{c^{EI}} log\left(\frac{1 - r^E}{r^E}\right) + \frac{G^E c^{EE} r^E + G^E I^{ext} - \mu^E}{c^{EI}} - \frac{1}{1 + exp(-(c^{IE} r^E - \mu^I)/\sigma^I)} = 0 \tag{5}$$

Due to the highly non-linear character of the equation, it is not possible to find a closed-form solution. Therefore, we find the fixed points by evaluating Eq (5) for all values of $r^E$ between 0 and 1, with a step of $10^{-6}$, and detecting the points $r^E_{fixed}$ where the function crosses 0 (Fig 1B). Then, the corresponding values of $r^I_{fixed}$ can be obtained by substituting $r^E$ by $r^E_{fixed}$ in Eq (4).

## Analytical approach to E-I homeostasis: Computation of model parameters at the fixed-point

To approximate the effect of E-I homeostasis in cortical networks, the parameters of the WC model should be adapted to maintain the activity of excitatory populations at a target level, regardless of perturbations in other parameters [17, 26, 48]. Here, to evaluate the conditions under which E-I homeostasis can maintain activity stable, we focus on changes at the level of the external input $I^{ext}$, as is commonly done in studies of E-I homeostasis through the use of sensory deprivation [26, 48]. Here, we explore four different modes of homeostasis derived

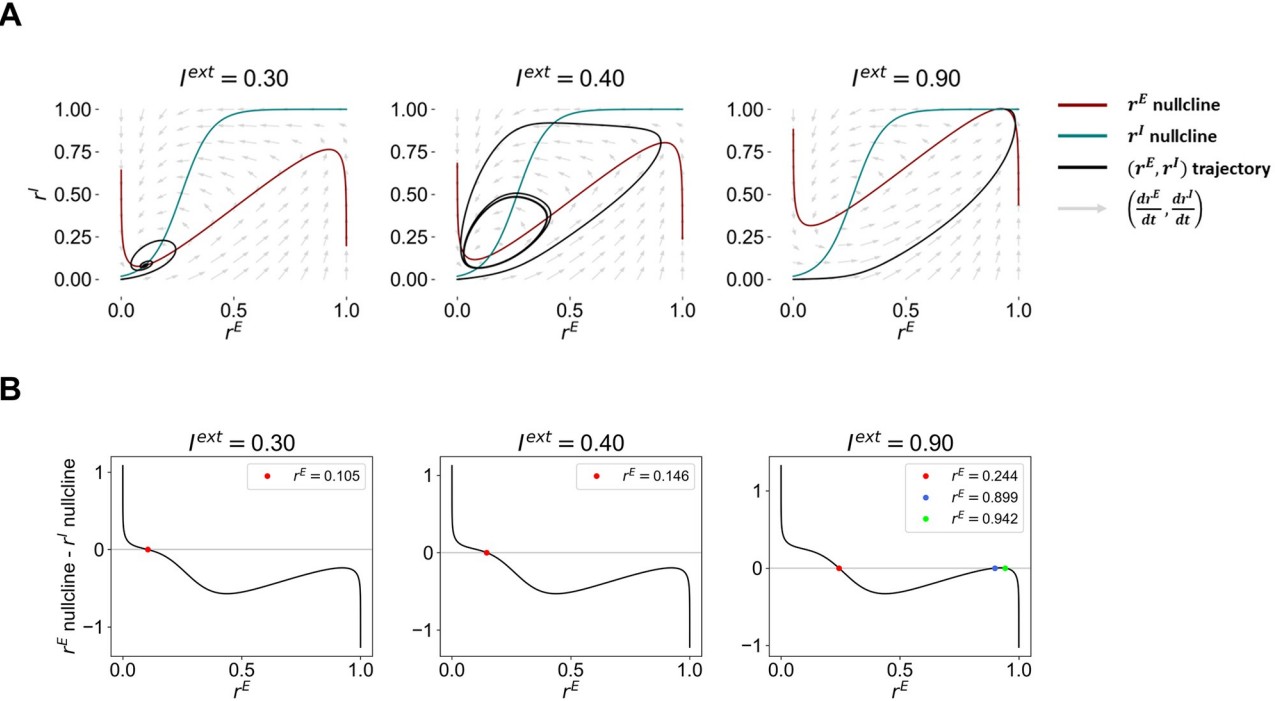

**Fig 1. Phase portraits of the Wilson-Cowan model and detection of fixed points.** (A) Phase portraits of the Wilson-Cowan model under different levels of external input $I^{ext}$. Red and blue lines represent the $r^E$ and $r^I$ nullclines, respectively, black lines represent the trajectory of the systems with initial condition (0, 0), and arrows represent the flux dictated by $\frac{dr^E}{dt}$ and $\frac{dr^I}{dt}$. (B) Detection of fixed points of the Wilson-Cowan model under different levels of external input $I^{ext}$.

from empirical studies (Fig 2) and, for each, we derive the expressions for obtaining the values of the parameters in question as a function of $I^{ext}$ and the target excitatory activity $r^E_{fixed}$.

Starting with synaptic scaling of excitatory synapses, it is known that, following perturbations in the normal firing rates of PY cells, their excitatory synapses are uniformly scaled to return firing rates to pre-perturbation levels [19, 21] through post-synaptic changes in excitatory receptors or synaptic spines [22, 47, 48]. In our implementation of the WC model, the strength of excitatory synapses on pyramidal populations can be scaled by adapting the parameter $G^E$ (Fig 2). For a given combination of $r^E$ and $I^{ext}$, we can then derive the homeostatic

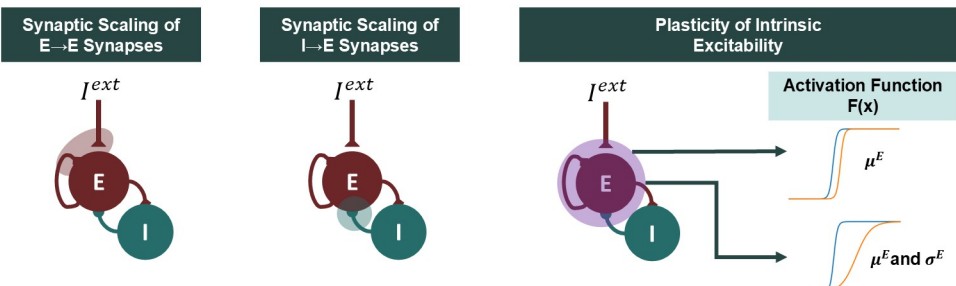

**Fig 2. Different implementations of excitatory-inhibitory homeostasis in the Wilson-Cowan model.** For each mode of homeostasis, we present a diagram of the Wilson-Cowan model, on which we highlight the model components that are modulated by E-I homeostasis. For plasticity of intrinsic excitability, we explore two methods which either modulate the threshold ($\mu^E$) of the input-output function, or adapt its threshold and slope ($\sigma^E$) in a coordinated manner.

solution of the system—the value of $G^E$ necessary to ensure that $r^E$ corresponds to a fixed point under $I^{ext}$—by isolating $G^E$ in Eq (5):

$$\frac{\sigma^E}{c^{EI}} log\left(\frac{1-r^E}{r^E}\right) + \frac{G^E c^{EE} r^E + G^E I^{ext} - \mu^E}{c^{EI}} - \frac{1}{1 + exp(-(c^{IE} r^E - \mu^I)/\sigma^I)} = 0$$

$$\frac{G^E(c^{EE} r^E + I^{ext})}{c^{EI}} = \frac{1}{1 + exp(-(c^{IE} r^E - \mu^I)/\sigma^I)} - \frac{\sigma^E}{c^{EI}} log\left(\frac{1-r^E}{r^E}\right) + \frac{\mu^E}{c^{EI}} \qquad (6)$$

$$G^E = \frac{1}{c^{EE} r^E + I^{ext}} \left(\frac{c^{EI}}{1 + exp(-(c^{IE} r^E - \mu^I)/\sigma^I)} - \sigma^E log\left(\frac{1-r^E}{r^E}\right) + \mu^E\right)$$

Importantly, following Dale's principle [58], we constrain $G^E$ so that it cannot become lower than 0, which effectively would mean that excitatory synapses become inhibitory.

Another common form of E-I homeostasis is the synaptic scaling of fast-spiking inhibitory synapses onto pyramidal neurons [12, 17, 21] (Fig 2, middle panel). Although it is not clear if synapses are adjusted in a pre- or post-synaptic manner [17] or if there is an overall reduction in the number of synapses [21, 22, 59], the function of this type of plasticity is also to maintain the firing rates of pyramidal neurons [17]. Similarly to $G^E$, the equation for the homeostatic value of $c^{EI}$, representing the inhibitory-to-excitatory coupling in the WC model, can be derived from Eq (5):

$$c^{EI} = \left(\sigma^E log\left(\frac{1-r^E}{r^E}\right) + G^E c^{EE} r^E + G^E I^{ext} - \mu^E\right)(1 + exp(-(c^{IE} r^E - \mu^I)/\sigma^I)) \qquad (7)$$

Similarly to $G^E$, we constrain $c^{EI} > 0$ to conform with Dale's principle.

Beyond synaptic scaling, there are mechanisms of E-I homeostasis that adapt the intrinsic excitability of PY neurons to counteract perturbations in firing rates, such as sensory deprivation or activity blockade [20, 23–25]. While some studies suggest that homeostasis of intrinsic excitability modulates firing thresholds [20] (Fig 2), others show, instead, concurrent modulation of the firing threshold and slope of input-output curves, so that the activity of a neuron with no external input remains the same [23–25] (Fig 2). Regardless, it is not clear to which extent cortical networks rely on these mechanisms of homeostasis [21], with the possibility that they only come into play following substantial perturbations in neuronal activity [17] or during development [25]. That said, homeostasis of intrinsic excitability can be implemented in two manners in the WC model. The first is at the level of the firing threshold of excitatory populations $\mu^E$ [20] (Fig 2). In this case, reordering Eq (2) gives:

$$\mu^E = \sigma^E log\left(\frac{1-r^E}{r^E}\right) - \frac{c^{EI}}{1 + exp(-(c^{IE} r^E - \mu^I)/\sigma^I)} + G^E c^{EE} r^E + G^E I^{ext} \qquad (8)$$

In the second case, the implementation of E-I homeostasis should ensure that both the firing threshold $\mu^E$ and slope $\sigma^E$ change in a coordinated manner so that $F^E(0)$ remains the same, in line with empirical results [23–25] (Fig 2 and S2 Fig). Starting with the default parameters

for the uncoupled node (Table 1) ($\mu_0^E = 1$ and $\sigma_0^E = 0.25$), we have:

$$F_0^E(0) = F^E(0)$$

$$\frac{1}{1 + e^{\frac{\mu_0^E}{\sigma_0^E}}} = \frac{1}{1 + e^{\frac{\mu^E}{\sigma^E}}}$$

$$\frac{\mu_0^E}{\sigma_0^E} = \frac{\mu^E}{\sigma^E}$$

$$\sigma^E = \frac{\sigma_0^E}{\mu_0^E}\mu^E = K\mu^E$$

(9)

where $K$ is a constant equal to $\frac{\sigma_0^E}{\mu_0^E}$. By substituting $\sigma^E$ for $K\mu^E$ in Eq (8) and isolating $\mu^E$, we obtain the expression for this mode of homeostasis as:

$$\mu^E = \frac{1}{K log\left(\frac{1-r^E}{r^E}\right) - 1}\left(\frac{c^{EI}}{1 + exp(-(c^{IE}r^E - \mu^I)/\sigma^I)} - G^E c^{EE} r^E - G^E I^{ext}\right)$$

$$\sigma^E = K\mu^E$$

(10)

In this case, a negative slope $\sigma^E$ would signify that excitatory inputs decrease population activity, and vice-versa, implying, in practice, a violation of Dale's principle. Therefore, we also apply the constraint that $\sigma^E = K\mu^E > 0$.

In addition to these forms of homeostatic plasticity, we also explored the synaptic scaling of excitatory synapses in inhibitory interneurons [43, 60], through modulation of the $c^{IE}$ parameter. However, due to the lack of robustness to higher levels of external input and the scarce empirical evidence, this mode of homeostasis is not explored in detail in this paper. Nonetheless, the value of $c^{IE}$ and model dynamics under different combinations of $r^E$ and $I^{ext}$ in the WC model with homeostasis of excitatory-to-inhibitory synapses can be consulted in S3 Appendix and S3 Fig.

## Implementation of multiple modes of homeostasis

Evidence suggests that different modes of homeostasis contribute simultaneously to the maintenance of E-I balance in the human cortex [17, 18, 20, 21, 42]. Therefore, we explore the possibility of the mathematical derivation of the system's fixed points under more than one type of E-I homeostasis. From Eq (7), it follows that the system can have multiple solutions of $(G^E, c^{EI})$ that satisfy the condition of maintaining $r_{fixed}^E$ for a given $I^{ext}$. Here, we present a heuristic process to select a combination of $G^E$ and $c^{EI}$ relying on the assumption that both forms of homeostasis have similar timescales $\tau_{homeo}$, which is in line with empirical results [20, 21, 26]. First, we define the putative equations describing the dynamics of $G^E$ and $c^{EI}$ as:

$$\tau_{homeo}\frac{dG^E}{dt} = -(r^E - \rho)$$

$$\tau_{homeo}\frac{dc^{EI}}{dt} = r^E - \rho$$

(11)

where $\tau_{homeo}$ is substantially larger than the time constants $\tau^E$ and $\tau^I$, representing the slow timescale of E-I homeostasis [17, 18] and ensuring separation of timescales between neural dynamics and homeostasis. Note that, when the firing rate $r^E$ is higher than the target $\rho$, $G^E$

will tend to decrease and $c^{EI}$ will tend to increase to bring firing rates toward lower levels, and *vice-versa*. Given that the magnitude of both derivatives is the same, $G^E$ and $c^{EI}$ have the same rate of variation at any point in time, albeit in opposite directions. Therefore, the absolute difference between steady-state $G^E$ and $c^{EI}$ and their initial values is always the same. For a more detailed mathematical proof and examples, refer to S4 Appendix and S4 Fig. Note that, as opposed to the common implementation of $c^{EI}$ homeostasis ($\frac{dc^{EI}}{dt} = r^I(r^E - \rho)$) [14, 36, 37, 39], our mechanism does not depend on $r^I$. This is relevant because, in this case, the assumption of equal variation from initial values would no longer hold. However, in S5 Appendix and S5 Fig, we provide a detailed explanation as to why our implementation of $c^{EI}$ homeostasis is more sensible for this version of the WC model. That said, under the assumption of equal timescales for the plasticity of $G^E$ and $c^{EI}$, we arrive at the following expression relating the steady state values of both parameters, accounting for the fact that they change in opposite directions:

$$
\begin{aligned}
G^E - G_0^E \quad &= -(c^{EI} - c_0^{EI}) \\
G^E \quad &= G_0^E - c^{EI} + c_0^{EI}
\end{aligned}
\tag{12}
$$

Therefore, by inserting this expression into Eq (7), substituting $G^E$ and reorganizing the terms, we arrive at the following equations for $c^{EI}$ and $G^E$ under synaptic scaling of both excitation and inhibition:

$$
\begin{aligned}
c^{EI} \quad &= \frac{\sigma^E log\left(\frac{1-r^E}{r^E}\right) + (G_0^E + c_0^{EI})(c^{EE}r^E + I^{ext}) - \mu^E}{\frac{1}{1+exp(-(c^{IE}r^E - \mu^I)/\sigma^I)} + c^{EE}r^E + I^{ext}} \\
G^E \quad &= G_0^E - c^{EI} + c_0^{EI}
\end{aligned}
\tag{13}
$$

where $G_0^E$ and $c_0^{EI}$ represent the initial values of $G^E$ and $c^{EI}$, corresponding to the default parameters of the WC model, presented in Table 1. Furthermore, in S4 Appendix and S4 Fig, we present an adaptation of our approach to accommodate different timescales for each homeostatic mechanism.

Assuming that homeostasis of intrinsic excitability operates in a timescale similar to the other homeostatic mechanisms, it is possible to derive an analytical expression for all the parameters under simultaneous $G^E$, $c^{EI}$, and $\mu^E$ homeostasis. In this case, we consider that:

$$
\tau_{homeo}\frac{d\mu^E}{dt} = r^E - \rho
\tag{14}
$$

In simpler terms, if firing rates are above the target, the threshold of the activation function increases to adjust activity toward the target setpoint. By combining this expression with Eq (11), we arrive at:

$$
\begin{aligned}
G^E \quad &= G_0^E - \mu^E + \mu_0^E \\
c^{EI} \quad &= c_0^{EI} + \mu^E - \mu_0^E
\end{aligned}
\tag{15}
$$

Then, we can combine Eq (15) with Eq (8), yielding:

$$
\mu^E = \frac{\sigma^E log\left(\frac{1-r^E}{r^E}\right) - \frac{c_0^{EI} - \mu_0^E}{1+exp(-(c^{IE}r^E - \mu^I)/\sigma^I)} + (G_0^E + \mu_0^E)(c^{EE}r^E + I^{ext})}{1 + \frac{1}{1+exp(-(c^{IE}r^E - \mu^I)/\sigma^I)} + c^{EE}r^E + I^{ext}}
\tag{16}
$$

Similarly, for plasticity of $\mu^E$ and $\sigma^E$, we substitute (15) in (10), obtaining:

$$\mu^E = \frac{\frac{c_0^{EI}-\mu_0^E}{1+exp(-(c^{IE}r^E-\mu^I)/\sigma^I)} - (G_0^E + \mu_0^E)(c^{EE}r^E + I^{ext})}{Klog\left(\frac{1-r^E}{r^E}\right) - 1 - \frac{1}{1+exp(-(c^{IE}r^E-\mu^I)/\sigma^I)} - c^{EE}r^E - I^{ext}} \tag{17}$$

## Computation of homeostatic parameters as a function of the target firing rate

In cortical networks, homeostatic plasticity mechanisms regulate the average firing rates of excitatory neurons [17, 19, 20, 26, 47, 48]. In the WC model, when the solution of the system is a stable fixed point or a stable spiral, the average firing rate of the system is precisely equal to $r_{fixed}^E$. However, when the target state corresponds to a limit cycle, the average firing rate differs from $r_{fixed}^E$ (Fig 3). For this reason, it is necessary to derive a method to obtain the solution of the system as a function of the target firing rate $\rho$, instead of $r_{fixed}^E$. There is, however, no analytical method able to provide the mean value of a limit-cycle. Therefore, to estimate the mean firing rate $\rho$ corresponding to a given $r_{fixed}^E$, the equations must be solved numerically and the mean $r^E$ computed from the data.

That said, to obtain an estimate of the steady-state model parameters as a function of $\rho$ and $I^{ext}$, we apply the following iterative procedure, illustrated in Fig 3, with $\epsilon = 10^{-10}$:

1. Define upper and lower limits for $r_{fixed}^E$ ($r_{lower}^E = 0$ and $r_{upper}^E = 1$).

2. For $r_{fixed}^E = \frac{r_{upper}^E - r_{lower}^E}{2}$, compute the homeostatic parameters of the system under $I^{ext}$ using the equations derived in the previous sections.

3. Solve the system numerically for 5 seconds and compute the average firing rate $\langle r^E \rangle$ from the last 2 seconds of activity.

4. If $|\langle r^E \rangle - \rho| < \epsilon$, take $r_{fixed}^E$ as the fixed point corresponding to $\rho$ and save the homeostatic model parameters.

5. Else, if $\langle r^E \rangle > \rho$, restart the procedure from step 1, updating $r_{upper}^E = r_{fixed}^E$. Conversely, if $\langle r^E \rangle < \rho$, restart from step 1 with $r_{lower}^E = r_{fixed}^E$.

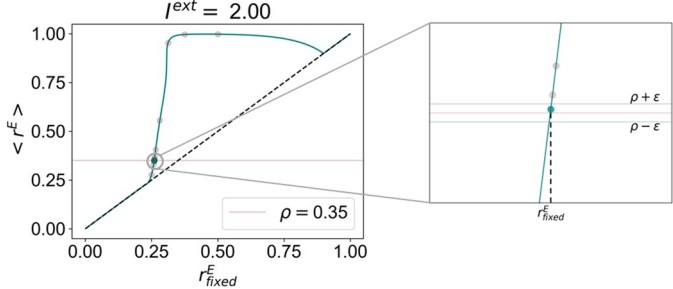

**Fig 3. Mean firing rate as a function of fixed point $r^E$ for the Wilson-Cowan model under homeostasis of $G^E$ and $I^{ext} = 2$.** While the solid blue line represents the mean firing rate as a function of $r_{fixed}^E$, the dashed black line corresponds to $\langle r^E \rangle = r_{fixed}^E$. Red dots show the iterations of the procedure. On the right, we include a zoom-in on the plot around the chosen value of $r_{fixed}^E$ corresponding to $\rho$.

This method allows for the iterative approximation of the model parameters corresponding to a mean firing rate of $\rho$, within an arbitrarily small level of precision dictated by $\epsilon$. Note that, for the lower values of $r_{fixed}^E$, for which the system solution corresponds to a stable fixed point, the mean firing rate is precisely the same as the fixed point $r_{fixed}^E$ (Fig 3). However, as the system enters the limit-cycle regime, the mean firing rate starts to grow faster than the fixed point $r_{fixed}^E$ and it is in this regime that the numerical estimation of parameters described above is necessary. Here, unless stated otherwise, the system was solved numerically using the Euler method with a time step of 0.1 ms.

## Derivation of Jacobian matrix and classification of system dynamics

After obtaining the fixed point solutions for each type of E-I homeostasis, we characterize network dynamics around the fixed points. The characterization of dynamics can be done with linear stability analysis [61, 62], by linearizing the system around a given fixed point using the first-order approximation of its Taylor expansion. To simplify the expressions, let us consider $x = \begin{pmatrix} r^E \\ r^I \end{pmatrix}$, $a = \begin{pmatrix} r_{fixed}^E \\ r_{fixed}^I \end{pmatrix}$, $f(x) = \frac{dr^E}{dt}$ and $g(x) = \frac{dr^I}{dt}$. With this notation, the first-order Taylor expansion of the WC system can be written as:

$$\frac{dx}{dt} = \begin{bmatrix} f(a) \\ g(a) \end{bmatrix}^{\!\!\!= 0} + \begin{bmatrix} \frac{df(a)}{dr^E} & \frac{df(a)}{dr^I} \\ \frac{dg(a)}{dr^E} & \frac{dg(a)}{dr^I} \end{bmatrix} (x - a) \tag{18}$$

The matrix containing the partial derivatives of $f(x)$ and $g(x)$ at the fixed point is the Jacobian ($J$) of the system and, through the analysis of the eigenvalues of $J$, one can evaluate the stability and dynamics of the system [61, 62]. A detailed derivation of the Jacobian in the WC model can be consulted in S6 Appendix. The eigenvalues $\lambda^{\pm}$ of $J$ can be expressed as a function of the trace ($Tr(J)$) and determinant ($Det(J)$) of the matrix as follows:

$$\lambda^{\pm} = \frac{Tr(J) \pm \sqrt{Tr(J)^2 - 4Det(J)}}{2}, \tag{19}$$

That said, the fixed point is stable if the real part of both eigenvalues of $J$ is negative (i.e. $Tr(J) < 0$ and $Det(J) > 0$) and the dynamics have periodicity if the eigenvalues have a non-zero imaginary component (i.e. $Tr(J)^2 - 4Det(J) < 0$). In addition, it is relevant to point out that fixed points in the "unstable spiral" regime do not necessarily represent an unstable solution of the system. As pointed out in [44], the WC model is always constrained by a bounding surface due to the sigmoid non-linearity, which, in our case, limits values between 0 and 1. Therefore, considering the Poincaré-Bendixson theorem [61], when there is only one fixed point, corresponding to an unstable spiral, and a boundary over which the flux points inwards, the system will necessarily settle on a periodic orbit.

Given the number of fixed points and the trace and determinant of $J$, we classify node dynamics in the following manner:

- **Stable Fixed Point**: only one fixed point, corresponding to $r_{fixed}^E$, $Tr(J) < 0$ and $0 < Det(J) < \frac{Tr(J)^2}{4}$.

- **Stable Spiral**: only one fixed point, corresponding to $r_{fixed}^E$, $Tr(J) < 0$ and $Det(J) > \frac{Tr(J)^2}{4}$.

- **Limit Cycle**: only one fixed point, corresponding to $r^E_{fixed}$, $Tr(J) > 0$ and $Det(J) > \frac{Tr(J)^2}{4}$.

- **Bistable**: two stable equilibria, one of which is the target $r^E_{fixed}$.

- **Unstable**: more than one fixed point with the target $r^E_{fixed}$ corresponding to an unstable solution of the system (unstable fixed point, unstable spiral, or saddle point).

In this study, we focus particularly on how different modes of E-I homeostasis modulate the Hopf-bifurcation between the stable spiral and limit cycle regimes, given the relevance of edge-of-bifurcation dynamics [27, 30–34], related to the bifurcation between stable activity and sustained oscillations [26–28].

### Analysis of intrinsic oscillation frequency

To analyze the impact of homeostasis on oscillation frequency, we solve the system numerically for each combination of $\rho$ and $I^{ext}$. Here, we simulate node dynamics for 5 seconds and use the last 3 seconds to compute the power spectrum of node activity, from which we extract the peak frequency of oscillation. To ensure that we can measure the frequency of systems in the stable spiral regime, which, if unperturbed, do not show rhythmic activity, we add a single pulse perturbation with amplitude 1 to $I^{ext}$ at 3 seconds of simulation time. While it is possible to estimate analytically the frequency of the WC model around a fixed point, this approximation is only valid in the vicinity of the fixed point, and when the system settles in a periodic orbit, the oscillation frequency decreases as the system orbits away from the fixed point (S7 Appendix and S6 Fig). For this reason, we analyze the effects of E-I homeostasis on oscillation frequency by solving the system numerically.

## Results

### Single and multiple modes of homeostasis in the Wilson-Cowan model

In cortical networks, E-I homeostasis has the function of maintaining stable firing rates in the face of perturbations in activity levels of PY neurons [17, 18]. In the experimental setting, such perturbations are often related to sensory deprivation [26, 48], which perturbs the magnitude of sensory input reaching early sensory cortices. While E-I homeostasis returns population firing rates to their average values pre-perturbation, it is not clear what is the exact value of such target firing rates, although studies suggest they correspond to the activity of cortical networks at the edge of bifurcation [26, 27]. For these reasons, we explore the response of the WC model relying on different modes of homeostasis, under different combinations of external input $I^{ext}$ and target firing rate $\rho$. To do that, we use the equations derived in the Methods section to estimate the steady state solution of different types of homeostasis as a function of $I^{ext}$ and $\rho$, together with the results of linear stability analysis around the fixed point (Fig 4). Here, for simplicity, we show results for all values of $\rho$ between 0 and 0.4, in steps of 0.005. However, we present model dynamics across values of $\rho$ from 0 to 1 in S7 Fig.

Our first conclusion is that single modes of homeostasis can be roughly divided into three groups, depending on how they shape the bifurcation between damped and sustained oscillations in the WC model. The first group includes $G^E$ homeostasis (Fig 4A) and the regulation of intrinsic excitability by modulation of $\mu^E$ and $\sigma^E$ (Fig 4D). For both, homeostasis modulates the value of $\rho$ where the Hopf bifurcation between damped and sustained oscillations is found. In this case, the bifurcation shifts toward higher values of $\rho$ as the external input is increased. For example, in the case of $\mu^E$ and $\sigma^E$ homeostasis, a target firing rate of $\rho = 0.12$ can correspond to limit-cycle oscillations, a stable spiral, or a stable fixed point. Therefore, models with

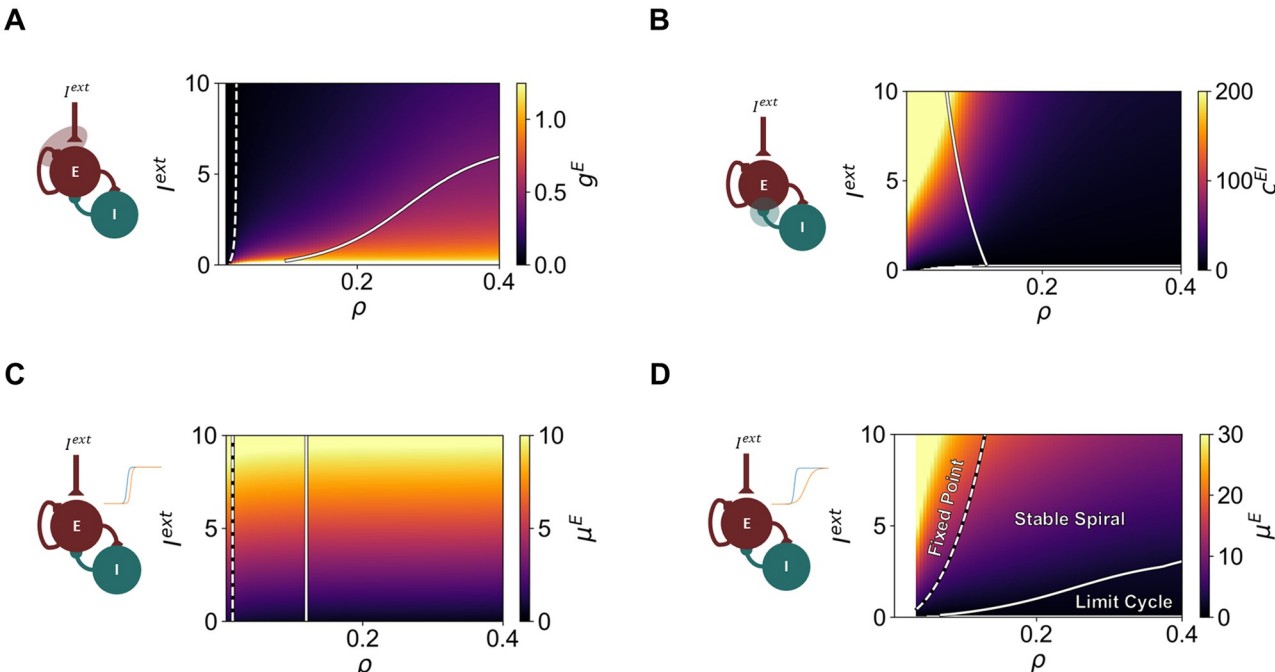

**Fig 4. Behavior of the Wilson-Cowan model under single modes of homeostasis as a function of $I^{ext}$ and target firing rate $\rho$.** (A) Homeostatic value of $G^E$ for different combinations of $\rho$ and $I^{ext}$ (B) Homeostatic value of $c^{EI}$ for different combinations of $\rho$ and $I^{ext}$ (C) Homeostatic value of $\mu^E$ for different combinations of $\rho$ and $I^{ext}$ under $\mu^E$ homeostasis (D) Homeostatic value of $\mu^E$ for different combinations of $\rho$ and $I^{ext}$ under coupled $\mu^E$ and $\sigma^E$ homeostasis. For all models, $\sigma^E = K\mu^E$. In all plots, dashed lines represent the transition from stable fixed point to stable spiral, while solid lines show the Andronov-Hopf bifurcation between a stable spiral and a limit cycle. The parameters were estimated using a timestep of 0.1 ms. For each mode of homeostasis, we include a diagram of the Wilson-Cowan model, indicating which model components are modulated by homeostatic plasticity. Blank areas relate to combinations of parameters for which the system has no solution (i.e. not possible to find a stable fixed point corresponding to $\rho$).

higher $I^{ext}$, under E-I homeostasis, are less likely to engage in sustained oscillations. The second group encompasses synaptic scaling of inhibition (Fig 4B), for which the bifurcation moves toward lower values of $\rho$ as the external input is increased. Therefore, for two nodes with the same target firing rate and different inputs, the one with stronger input might be in the limit-cycle regime, while its counterpart displays damped oscillations. Therefore, in this case, homeostatically regulated models with higher external input are more likely to enter the limit cycle regime. Finally, for the third group, consisting of $\mu^E$ homeostasis (Fig 4C), the bifurcation between the stable spiral and limit cycle regimes is only dependent on the target firing rate, occurring at $\rho \approx 0.12$. In this case, model dynamics can be regulated solely through the target firing rate.

Furthermore, our results also suggest that all four modes of homeostasis are robust to changes in $I^{ext}$ and $\rho$, being able to maintain activity close to the target firing rate, provided it is not too low (see, for example, blank space in Fig 4D). However, we note that, in models relying on the homeostasis of inhibition (Fig 4B), the modulations in model parameters required to ensure that excitatory firing rates are maintained at $\rho$ are orders of magnitude larger in comparison with the other modes of homeostasis. For example, for higher values of $I^{ext}$ and lower target firing rates ($\rho$), the value of $c^{EI}$ needs to be scaled to up to 250, which is two orders of magnitude larger than the default value of 2.5. As demonstrated in the following sections, this can have a strong impact on node dynamics, particularly the intrinsic oscillation frequency.

Beyond the single modes of homeostasis, we investigate the dynamics of the WC model under multiple modes of plasticity, and how they are shaped by external inputs and the target

firing rate $\rho$. The results for homeostasis of $G^E$ and $c^{EI}$ are presented in Fig 5A. Notably, homeostasis no longer requires substantial modulation of model parameters, particularly $c^{EI}$, which can minimize the impact of E-I homeostasis in model dynamics that might be caused by substantial variation in the model parameters [43] (S8 Fig). On another note, the modulation of the bifurcation point by homeostasis is similar to what is observed for homeostasis of $G^E$, shifting the bifurcation point toward higher values of $\rho$ when the external input is increased. This indicates that, when combining homeostasis of parameters with multiplicative (i.e $G^E$) and subtractive (i.e. $c^{EI}$) effects on model activity, the effects on dynamics will be dominated by the multiplicative parameters. This is already visible for homeostasis of $\mu^E$ and $\sigma^E$ (Fig 4D), where the parameter space is similarly dominated by the effect of changing $\sigma^E$.

Regarding the homeostasis of $G^E$, $c^{EI}$, and the excitability threshold $\mu^E$ (Fig 5B), we observe that the magnitude of variation of the model parameters is further reduced (S8 Fig). In addition, while the dominance of $G^E$ in shaping the parameter space is still present, the combined action of $c^{EI}$ and $\mu^E$ is apparent in the extension of the limit-cycle region to higher values of $I^{ext}$. Conversely, when plasticity of intrinsic excitability modulates $\mu^E$ and $\sigma^E$ [23, 24] (Fig 5C), $\sigma^E$ has the opposite effect, extending the stable spiral region in comparison with the model with $G^E$ and $c^{EI}$ homeostasis only. Finally, it is worth noting that the combination of plasticity of $\mu^E$ and $\sigma^E$ with other modes of homeostasis also makes it more robust to lower target firing rates (compare blank spaces in Figs 4D and 5C).

In conclusion, across all modes of homeostasis, the influence of multiplicative parameters such as $G^E$ and $\sigma^E$ dominates node dynamics, leading to a modulation of the bifurcation between the stable spiral and limit-cycle regimes, shifting it toward higher values of $\rho$ when $I^{ext}$ increases. Conversely, for $c^{EI}$ homeostasis, the opposite effect occurs. Finally, for $\mu^E$ homeostasis, our results suggest that the bifurcation only depends on $\rho$. Furthermore, when combining multiple modes of homeostasis, the effect of multiplicative parameters dominates, shifting the Hopf bifurcation to higher values of $\rho$ as the external input is increased. On another note, the combination of multiple modes of homeostasis contributes to reducing the magnitude of parameter variations, which can be particularly strong for the plasticity of inhibitory synapses.

## Effect of E-I homeostasis on the natural frequency of oscillations

The generation of rhythmic activity in the WC model depends on the push-pull between excitation and inhibition. Therefore, given that E-I homeostasis can modulate the strength of excitation and inhibition, it is relevant to explore the effect of homeostasis on the frequency of oscillation. Following the procedure introduced in the methods section, we analyze how the intrinsic frequency of oscillation of the WC model changes based on the combination of $\rho$ and $I^{ext}$, for different implementations of E-I homeostasis (Fig 6A).

Starting with $G^E$ homeostasis, we observe that, in the stable spiral regime, the frequency of oscillation is largely independent of $I^{ext}$ and increases up to $\rho \sim 0.27$, where it reaches a peak close to 125 Hz, after which the frequency starts decreasing with $\rho$. As nodes enter the limit cycle, the frequency of oscillation also decreases with $\rho$, but to a lesser extent. In addition, for values of $\rho$ close to 0.1, the frequency of oscillation corresponds to the low gamma band (30–60 Hz). Conversely, $c^{EI}$ homeostasis has a strong impact on the frequency of oscillations, which become particularly fast around the Hopf bifurcation. In this region of the parameter space, the model can display oscillations of up to 250 Hz due to the high magnitude of inhibition required to maintain activity at the target firing rate (Fig 4B). Since oscillations are generated through the reciprocal interactions between excitatory and inhibitory populations, increasing $c^{EI}$ has a pronounced impact on the frequency of oscillations. Accordingly, both forms of homeostasis of intrinsic excitability do not have such a strong effect on the frequency

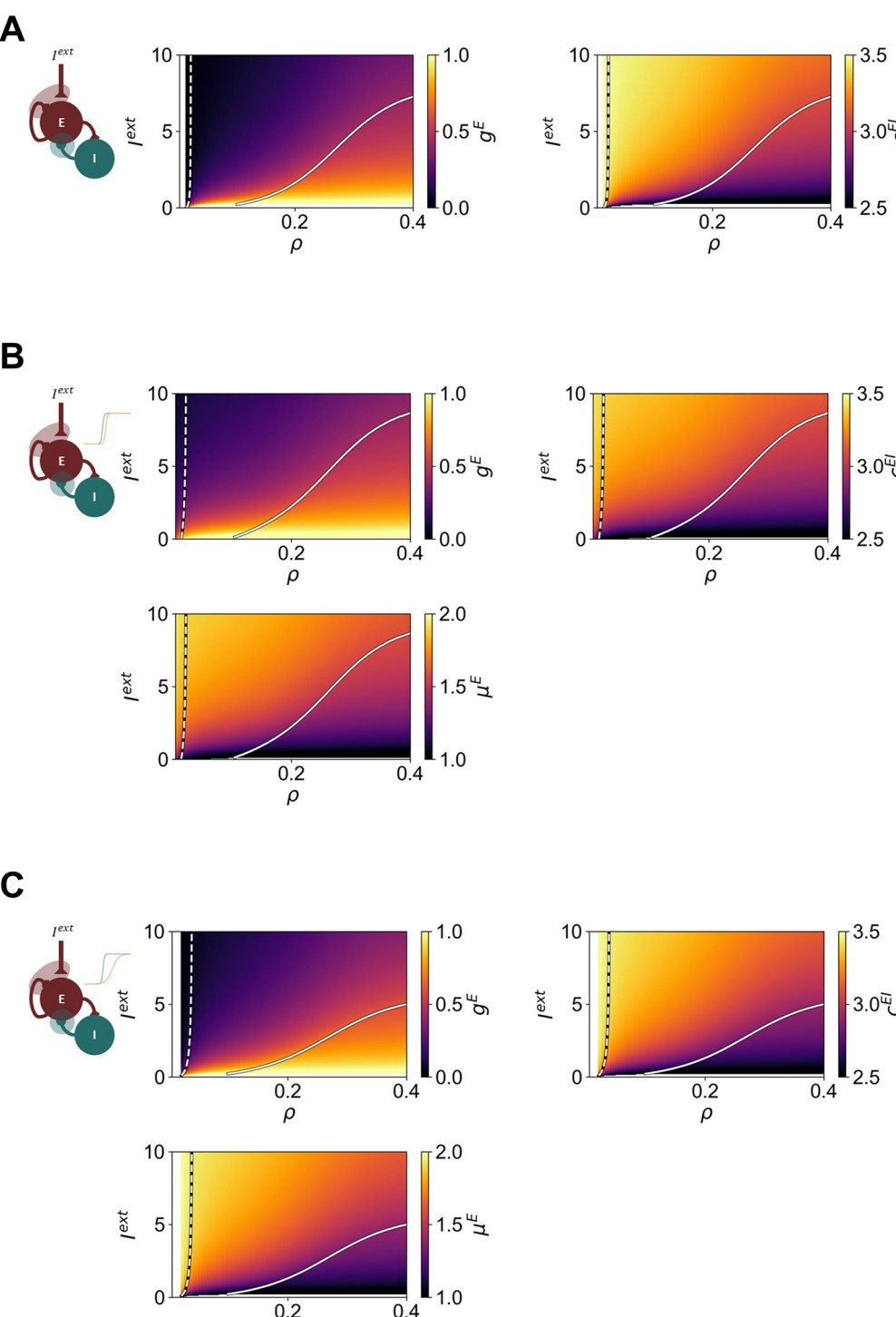

**Fig 5. Behavior of the Wilson-Cowan model under multiple modes of homeostasis as a function of $I^{ext}$ and target firing rate $\rho$.** (A) Homeostatic values of $G^E$ (Left) and $c^{EI}$ (Right) (B) Homeostatic values of $G^E$ (Top left), $c^{EI}$ (Top right) and $\mu^E$ (Bottom) under homeostasis of $G^E$, $c^{EI}$ and $\mu^E$ (C) Homeostatic values of $G^E$ (Top left), $c^{EI}$ (Top right) and $\mu^E$ (Bottom) under homeostasis of $G^E$, $c^{EI}$, $\mu^E$ and $\sigma^E$, with $\sigma^E = K\mu^E$. In all plots, dashed lines represent the transition from stable fixed point to stable spiral, while solid lines show the Andronov-Hopf bifurcation between a stable spiral and a limit cycle. The parameters were estimated using a timestep of 0.1 ms. For each combination of modes, we include a diagram of the Wilson-Cowan model, indicating which model components are modulated by homeostatic plasticity. Blank areas relate to combinations of parameters for which the system has no solution (i.e. not possible to find a stable fixed point corresponding to $\rho$).

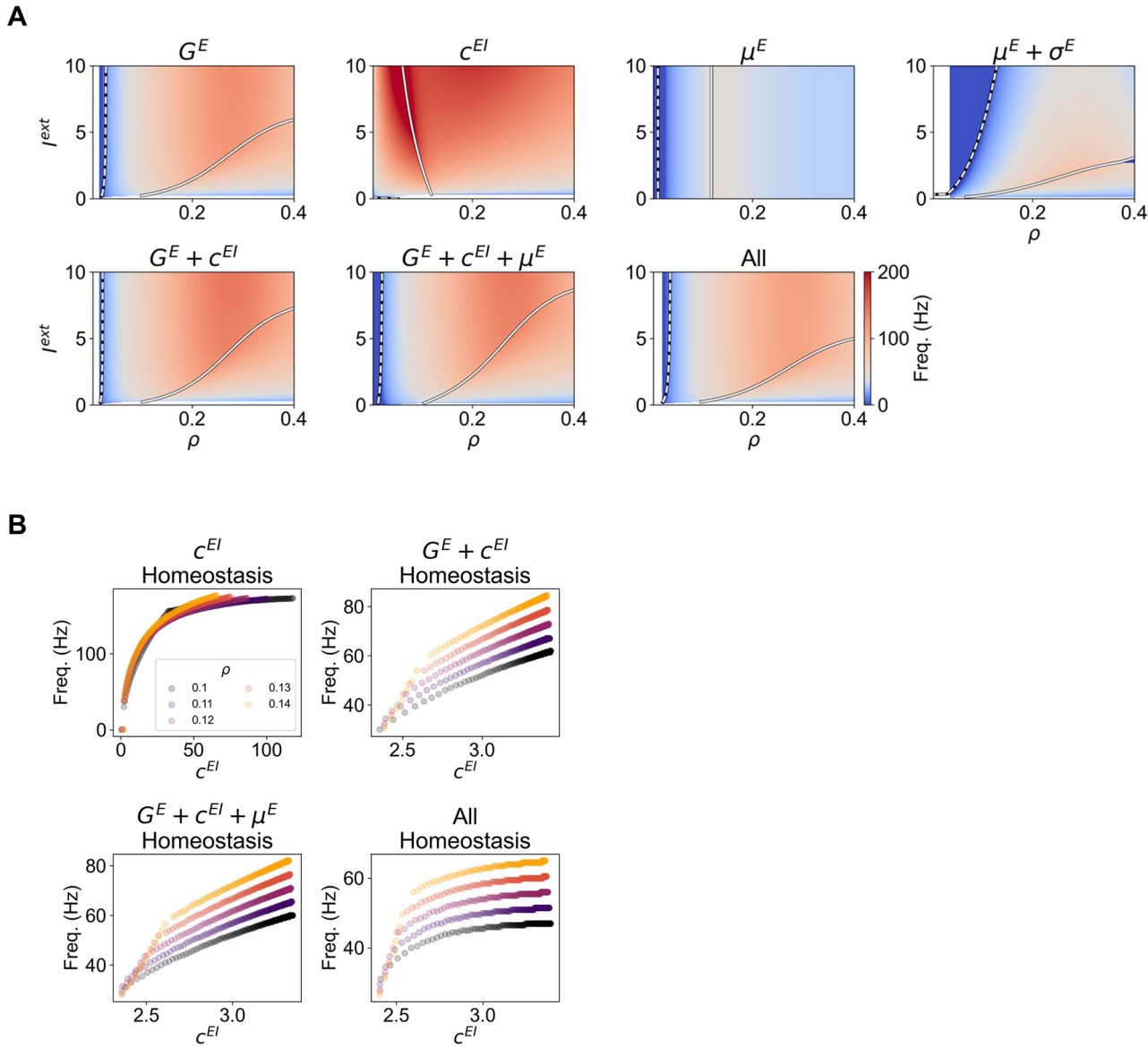

**Fig 6. Effect of E-I homeostasis on oscillation frequency.** (A) Peak frequency of oscillation of the Wilson-Cowan Model under Different Modes of Homeostasis as a Function of $I^{ext}$ and target firing rate $\rho$. We present the peak frequency of oscillation across the parameter space for all modes of homeostasis. The colormap is centered around 40 Hz, the frequency of oscillation of the default model. In addition, we also display the bifurcations between the stable fixed point and stable spiral regimes (dashed line) and between the stable spiral and limit cycle (solid line). (B) Oscillation frequency as a function of $c^{EI}$ for different modes of homeostasis. We present the relationship between the intrinsic frequency of oscillation and the strength of inhibition $c^{EI}$ in models with $c^{EI}$ (Top Left), $G^E + c^{EI}$ (Top Right), $G^E + c^{EI} + \mu^E$ (Bottom Left) and $G^E + c^{EI} + \mu^E + \sigma^E$ (Bottom Right) homeostasis. In addition, we plot the relationship in models with different target firing rates ($\rho$), ranging from 0.1 to 0.14. In all plots, each dot represents a model with a different level of incoming input $I^{ext}$.

of oscillation. For example, for $\mu^E$ homeostasis, the frequency only depends on $\rho$, peaking close to the Hopf bifurcation at $\sim 45$ Hz. Conversely, coupled homeostasis of $\mu^E$ and $\sigma^E$ impacts the frequency of oscillation, since $\sigma^E$ not only scales the external input but also the strength of the excitatory and inhibitory inputs to pyramidal populations. Therefore, this type of homeostasis modulates the frequency of oscillations more strongly than the adaption of $\mu^E$ only. Nonetheless, since scaling $\sigma^E$ has the same effect on the inhibitory and excitatory couplings, the impact

on the frequency is not as strong as synaptic scaling of either $G^E$ and $c^{EI}$. In addition, similarly to $c^{EI}$ and $\mu^E$ homeostasis, the frequency of oscillation peaks around the Hopf bifurcation.

In addition to the impact of single modes of homeostasis, we further present how the conjugation of multiple modes shapes oscillations in the WC model (Fig 6A). Starting with $G^E$ and $c^{EI}$ homeostasis, since $c^{EI}$ does not vary as strongly as for $c^{EI}$ homeostasis (see Figs 4B and 5A), the frequency remains closer to 40 Hz in a broader region of the parameter space, compared to $c^{EI}$ homeostasis, even though it still peaks at higher values. Here, the dominance of multiplicative parameters such as $G^E$ is further suggested by the similarity between the parameter spaces of $G^E$ and $G^E + c^{EI}$ homeostasis. Furthermore, while the inclusion of $\mu^E$ homeostasis has no substantial effects compared to the homeostasis of $G^E$ and $c^{EI}$ only, the implementation of plasticity of intrinsic excitability with $\mu^E$ and $\sigma^E$ reduces the impact of homeostatic plasticity on the frequency of oscillation, which remains in the low-gamma range in a larger region of the explored parameter space.

On another note, studies have demonstrated that the frequency of gamma oscillations in different areas of the human cortex correlates significantly with the density of inhibitory receptors [63, 64]. Importantly, although this relationship is found, the frequency of physiological low-gamma oscillations remains roughly bounded between 30 and 100 Hz. To investigate if this interaction is also found in our neural-mass model, we plot the peak frequency of gamma oscillations against the inhibitory coupling $c^{EI}$ in all models involving the homeostasis of inhibition (Fig 6B). Our results demonstrate that the empirical relationship between inhibition and gamma frequencies is a general feature of our model, regardless of the combination of homeostatic parameters or the target firing rate. However, as previously pointed out, in models with the plasticity of inhibition only (Fig 6B), the frequency goes beyond the range reported in [63, 64]. Conversely, when relying on multiple homeostatic mechanisms, cortical circuits maintain gamma rhythms within physiological bounds (Fig 6B), further showcasing the circuit-level benefits of combining multiple mechanisms for firing-rate homeostasis.

All in all, we demonstrate that, beyond the maintenance of target firing rates, different modes of homeostasis can have profound impacts on node dynamics such as the frequency of oscillations. Importantly, this result has implications for the oscillatory activity in cortical areas with different densities of excitatory and inhibitory receptors, which might arise due to the effect of E-I homeostasis. Therefore, we stress that this effect should be considered when interpreting results showing a relationship between gamma oscillation frequency and the density of inhibitory receptors in the cortex.

## Effect of integration time step (dt) on node dynamics

The method derived to estimate the homeostatic values of model parameters as a function of the target firing rate $\rho$ requires the numerical integration of the WC model. Therefore, from a methodological standpoint, it is relevant to investigate the impact of the integration time step ($dt$) on the results. More specifically, the rhythmic activity in the WC model is a consequence of the interplay between excitation and inhibition, leading to a constant push-pull that generates oscillations that might be damped or sustained. That said, given that the integration time step can influence how fast inhibition and excitation evolve, the behavior of the model can be affected, shifting the position of the bifurcation. For this reason, we analyze the position of the Andronov-Hopf bifurcation as a function of $\rho$ and $I^{ext}$ and how it varies depending on the level of external input.

Our results suggest that, for all modes of homeostasis and independently of $I^{ext}$, the position Hopf-Bifurcation is affected by changing the integration $dt$ (Fig 7). Some models are minimally impacted, particularly when relying on either type of plasticity of intrinsic excitability,

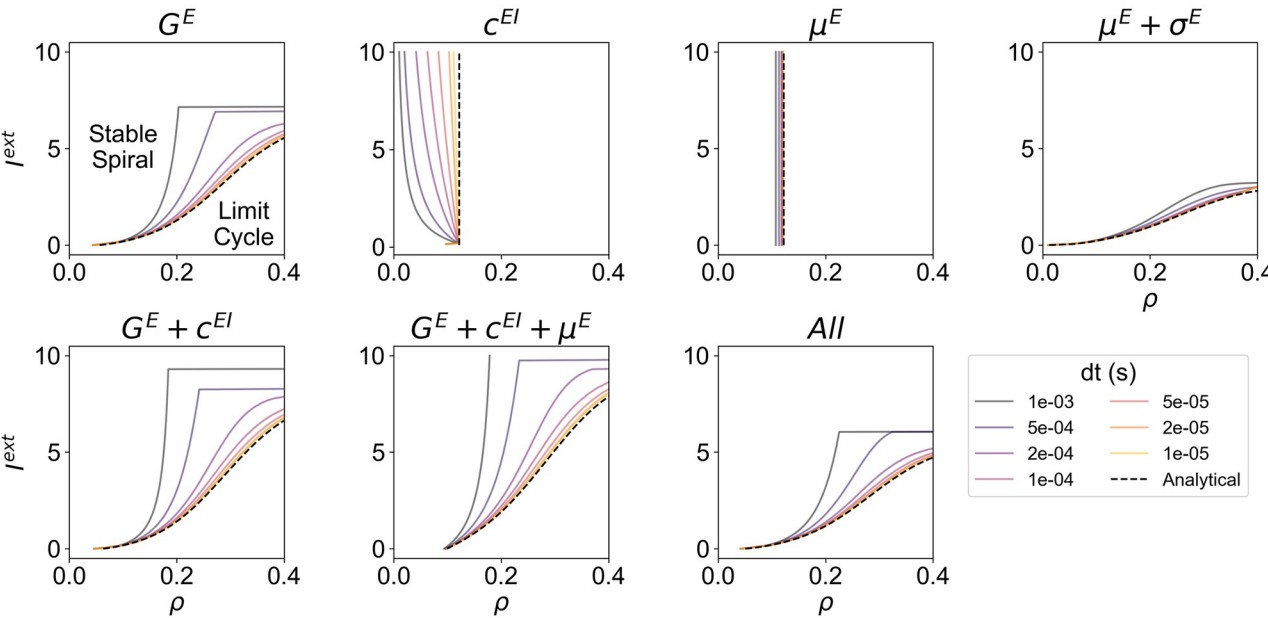

**Fig 7. Hopf-Bifurcation as a function of the integration time step (*dt*).** For each mode of homeostasis, dashed black lines represent the analytical bifurcation between the stable spiral and limit cycle regimes as a function of $\rho$ and $I^{ext}$. In addition, we present the bifurcation lines computed numerically using the Euler method with different integration time steps, ranging from $10^{-5}$ to $10^{-3}$ seconds.

likely because these modes of homeostasis do not directly affect the E-I interactions in the model. Conversely, particularly in modes of homeostasis relying on the plasticity of $c^{EI}$, the integration *dt* has a strong influence on the bifurcation, with slower *dt* values extending the limit-cycle regime to lower values of $\rho$.

Importantly, when analyzing the analytical bifurcation portraits, it can be observed that $c^{EI}$ homeostasis can be more accurately grouped with $\mu^E$ homeostasis as types of plasticity for which the bifurcation depends only on $\rho$. However, when integrating the system numerically, we observe the behavior reported in the previous section, where the bifurcation shifts toward lower values of $\rho$ with increases in $I^{ext}$. The existence of two groups of homeostatic mechanisms can be explained by the way each parameter shapes the external input in the WC model (Eq 1). More specifically, as previously hinted at, $c^{EI}$ and $\mu^E$ have a subtracting effect on the external input (direct in the case of $\mu^E$ and indirect in the case of $c^{EI}$), while $G^E$ and $\sigma^E$ have a multiplicative relation with $I^{ext}$. This results in the different dependence of dynamics on $\rho$ and $I^{ext}$. While here we focus on the dynamics of isolated WC neural masses, the implications of these differences on the network context should be subject to further study, especially regarding network stability.

That said, since large-scale models are often solved numerically and given the non-negligible effect of integration *dt*, our results suggest that, when analyzing the bifurcation dynamics of the WC model, especially in the network context, the effect of *dt* should be accounted for and minimized. Furthermore, from the inspection of our results, values of *dt* lower than 0.2 milliseconds generally provide a sufficient approximation of the analytical behavior of the model while maintaining simulation times tractable. While this might not be the case for the homeostasis of $c^{EI}$, where integration invariably affects the independence of the bifurcation on $I^{ext}$, we suggest that the numerical bifurcation portrait, as opposed to the analytical one, should be considered when studying the dynamics of models relying only on $c^{EI}$ homeostasis.

### Bifurcation control in models with fast and slow inhibition

As stated in the previous section, rhythmic activity in cortical networks can be a consequence of the interplay between excitation and inhibition, leading to a constant push-pull that generates oscillations that might be damped or sustained. Our default WC model is based on the generation of gamma rhythms ($\sim 40$ Hz) by the PY-PV loop, due to the fast GABAergic synapses that fast-spiking PV interneurons establish in PY neurons [52, 53, 55]. However, other classes of inhibitory interneurons modulate pyramidal activity through $GABA_B$ receptors, which have slower dynamics [51, 65, 66]. Furthermore, this can lead to the resonance at rhythms slower than gamma (i.e. alpha/beta), as observed in the deeper layers of the cortex [67–70]. That said, E-I homeostasis may shape the dynamics of cortical networks differently depending on the time constants of inhibitory synapses. Therefore, we explore how the time constant of inhibition ($\tau^I$) interacts with the modulation of the Andronov-Hopf bifurcation by each mode of homeostasis (Fig 8).

Our results indicate that, even though each mode of homeostasis displays a different bifurcation portrait, a common effect of increasing the inhibitory time constant can be discerned: as inhibition becomes slower, the bifurcation shifts toward lower values of $\rho$. Furthermore, while this effect is common across all modes of homeostasis, it is less strong in the model relying on the plasticity of intrinsic excitability through the threshold ($\mu^E$) and slope ($\sigma^E$) of the input-output function (see $\mu^E + \sigma^E$ panel in Fig 8). The main implication of our result is that, generally and regardless of the implementation of E-I homeostasis, cortical networks with slower inhibition might be more likely to engage in sustained oscillations.

In summary, our results suggest that the critical point between damped and sustained oscillations is shaped not only by the mechanisms of E-I homeostasis but also by the time constant of inhibition. More specifically, for a given level of input $I^{ext}$, slower inhibition shifts the bifurcation to lower target firing rates ($\rho$). The implications of this behavior in cortical networks are discussed in detail in the Discussion.

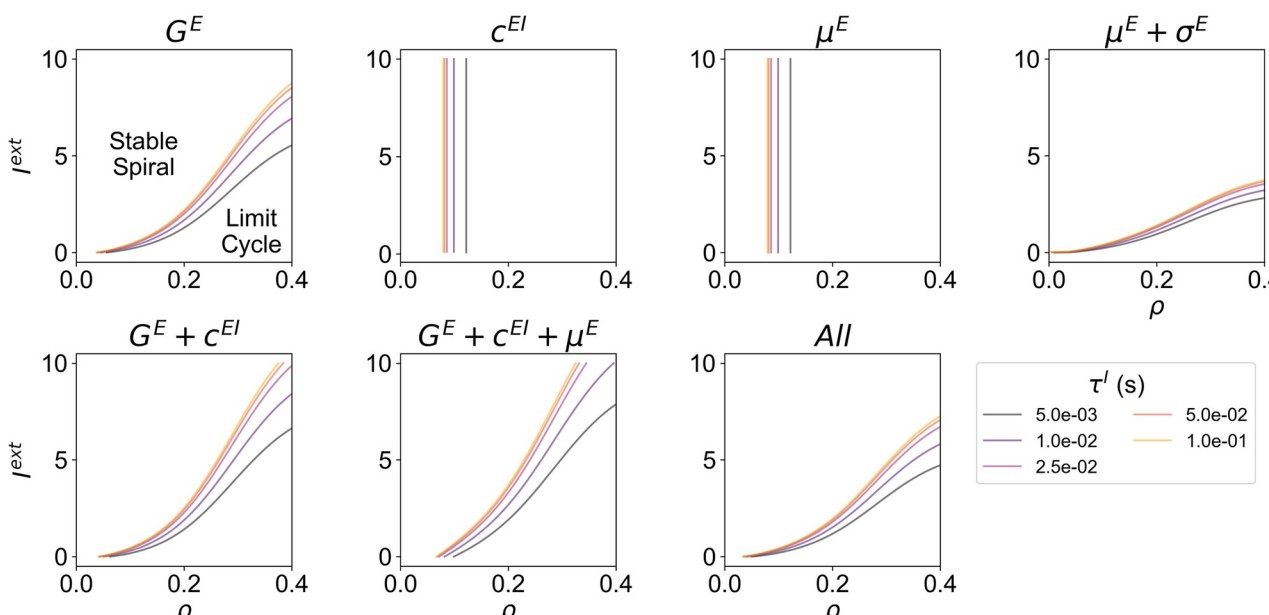

**Fig 8. Hopf-Bifurcation as a function of the time constant of the inhibitory population ($\tau^I$).** For each mode of homeostasis, we present the analytical Hopf-bifurcation as a function of $\rho$ and $I^{ext}$, in models with $\tau^I$ ranging from 5 to 100 ms. In all simulations, the excitatory time constant ($\tau^E$) was set to 2.5 ms.

## Discussion

In this work, we apply concepts from the field of dynamical systems, such as bifurcation theory and linear stability analysis, to explore the behavior of the Wilson-Cowan model under different modes of E-I homeostasis. To do this, we gather information from the literature on the mechanisms through which cortical networks regulate their balance to maintain stable firing rates [17, 18, 42] and translate them to equivalent adaptations in the parameters of the Wilson-Cowan model. This approach allows for the derivation of the model parameters that allow for systems to be poised at a given target firing rate $\rho$ under different levels of incoming input. In addition, to account for the concurrent action of multiple homeostatic mechanisms and study their synergies, we derive a method to estimate steady-state parameters of models under multiple modes of homeostasis, under the assumption that all modes operate with the same timescale.

One of the most prominent advantages of our approach is methodological and concerns the computational traceability of models with E-I homeostasis. Normally, the implementation of this form of plasticity in large-scale models requires that simulations are run for a substantial amount of simulation time, ensuring that model parameters reach a steady state (see, for example, [40, 41]). Therefore, being able to estimate the steady-state solution of E-I homeostasis analytically may allow for a more extensive exploration of E-I balance in large-scale models, which would require massive computational resources otherwise. Furthermore, within the topic of multiple modes of homeostasis, it is worth pointing out that, even though we assumed equality of timescales between different types of homeostasis to reach the expressions for the homeostatic parameters [17, 20], our mathematical framework allows for the estimation of steady-state parameters in models of different homeostatic timescales (S4 Appendix and S4 Fig). On the other hand, the homeostatic set-point of circuits with multiple homeostatic mechanisms is also dependent on the initial value of each parameter (S4 Appendix and S4 Fig). This raises the question of how much these values shape the steady-state dynamics of the system and what is the range of initial conditions tolerated by the circuit without detriment to its behavior. Therefore, we suggest that future computational studies should focus not only on this question, but also on fitting the default parameters of the model with population recordings from cortical slices, where the external input can be easily controlled. That said, we believe that our results serve to provide the mathematical framework to study these questions in the future.

Beyond the practical benefits of our approach, it also allows for a deeper understanding of the dynamics of the balanced Wilson-Cowan model and, consequently, of networks of coupled excitatory and inhibitory neurons in the cortex. For this reason, we explore the behavior of the model under homeostasis of excitatory-to-excitatory synapses, excitatory-to-inhibitory synapses, and intrinsic excitability, in addition to the most common implementation of homeostasis in inhibitory-to-excitatory connections [14, 36–39]. In this context, our bifurcation analysis suggests that the modes of homeostasis can be divided into two groups. The first, including plasticity of $c^{EI}$ and $\mu^E$, relates to models where the bifurcation between a fixed point solution and a limit cycle is only dependent on the target firing rate $\rho$ for most of the domain. In a network context, this would mean that all nodes could be poised in the same dynamical regime regardless of heterogeneities in external inputs. However, when simulating model activity by solving the system numerically, the integration $dt$ has a strong effect on models with $c^{EI}$ homeostasis. More specifically, the bifurcation is shifted toward lower values of $\rho$ as the external input is increased, meaning that nodes with higher inputs are more likely to be in the limit cycle regime. Conversely, in the other group, including plasticity of $G^E$ and coupled $\mu^E$ and $\sigma^E$, the bifurcation depends on both the external input and the target firing rate. Therefore, in the

network context, nodes can maintain the same firing rate while being in different regimes relative to the bifurcation. Regarding the advantages of each type of plasticity, there is no clear picture at this level of analysis. On one hand, research suggests that cortical networks self-regulate toward quasi-criticality by regulating firing rates [18, 26]. Under the assumption that cortical networks achieve this through the maintenance of a uniform set point firing rate, methods such as $\mu^E$ homeostasis would be at an advantage. On the other hand, the study of [26] relates to sensory areas and it is unclear if all areas of the cortex have the same optimization goal point relative to criticality. In this context, research suggests that optimal cortical dynamics occur when nodes have heterogeneous distances to the bifurcation [32]. This could be either achieved by having heterogeneous target firing rates or, in our model, through modes of homeostasis modulating multiplicative parameters (e.g. $G^E$ or $\sigma^E$), for which this heterogeneity emerges naturally from different levels of external input. Furthermore, recent results suggest that distance to criticality (equivalent to the distance to bifurcation) tracks the hierarchy of visual areas, in line with the idea of heterogeneity in distance to bifurcation across the cortex [35]. Within our framework, this could be a consequence of the interaction between different levels of input or heterogeneity in microcircuitry [7] and the effects of particular modes of homeostasis modulating the distance to the bifurcation. Therefore, future studies should focus on the comparison between the behavior of network models under the different modes of homeostasis, investigating how the different types of bifurcation portraits shape network dynamics. Nonetheless, our results agree with the edge-of-bifurcation theory of brain dynamics [27, 30, 31, 71, 72], demonstrating that bifurcation control can be implemented in cortical networks through homeostatic regulation of firing rates, following previous theoretical [27] and empirical studies [26]. More importantly, we offer a plausible explanation for the heterogeneity in the distance to bifurcation across cortical areas [32, 35], by suggesting that this can result from the interaction between E-I homeostasis and local heterogeneities in incoming inputs or microcircuitry [7], without requiring heterogeneous target firing rates. On another note, a relevant phenomenon in edge-of-bifurcation systems that was not studied here is critical slowing-down, relating to the longer time necessary to recover from perturbations in systems close to the bifurcation [73]. While, at the neuronal level, critical slowing-down is thought to provide relevant information about the transition between the quiescent and spiking regimes [74], its effects on brain dynamics at the circuit level are not as clear. Therefore, we suggest that our framework can be used in future studies to study critical slowing down near the Hopf-bifurcation and how it shapes the generation of cortical rhythms.

Here, we also study how each mode of homeostasis shapes the frequency of oscillations generated by the model. Recent research shows that, in the WC model, the frequency is modulated by changes in the coupling parameters and external input [46]. This results from the fact that oscillations in the WC model emerge from the bidirectional interaction between E and I neurons. Shortly, the E population excites itself and the I population, which leads to an increase in E and I activity, until the latter reaches a point where it can compensate for the recurrent excitation and external input. At this point, both E and I activities start decreasing until the I activity is no longer able to compensate for the recurrent excitation, which leads to a new cycle. Therefore, it is to be expected that changes in $G^E$ or $c^{EI}$, which regulate how quickly excitation and inhibition evolve, also lead to a modulation in the duration of cycles. For this reason, we study how homeostasis shapes the frequency of oscillations (Fig 6). For example, our results demonstrate that $c^{EI}$ homeostasis, which is the most common method implemented in large-scale models [36–41], has a profound effect on the oscillatory behavior of the model. This is because changes in the external input require large modulations of $c^{EI}$ which strongly affects the frequency of oscillation. For this reason, even though the target firing rate might be common, nodes with different external inputs will oscillate at different frequencies. Conversely, for

$G^E$ homeostasis, the frequency of oscillation has a smaller range of variation across different levels of input. While changing $G^E$ should modulate the frequency due to its effect on recurrent excitation, this is compensated by the modulation of external inputs by the same parameter. Therefore, through different types of homeostasis, nodes with varying inputs can have varying levels of heterogeneity in oscillation frequency. How does this relate to cortical dynamics? Results in the human cortex suggest that the frequency of gamma oscillators in the visual cortex correlates with the density of inhibitory receptors [63]. Furthermore, recent results show that this correlation extends to other cortical areas, possibly tracking the cortical hierarchy [64]. This is in line with our results regarding modes of homeostasis involving $c^{EI}$, where the frequency of oscillation increases with the inhibitory coupling (Fig 6B). However, our results show that, when implementing homeostasis of inhibition only, the frequency can vary up to 200 Hz, which is outside the range of variation reported in [63, 64] ($\sim$ 35–100 Hz). Importantly, this issue can be solved through the integration of multiple modes of homeostasis. In this context, our results demonstrate that, since the magnitude of variation in parameters is smaller, there is a weaker impact of homeostasis on the frequency of oscillation when multiple modes are integrated. For example, under combined homeostasis of $G^E$ and $c^{EI}$ (6B), we find that the frequency still varies with the level of external input, and thus the strength of inhibition, but it remains within the range reported by [63, 64]. This can be beneficial for long-range synchrony between cortical areas in the gamma range, which is essential for neural communication [75] and behavior [76]. That said, we argue that our results are in line with empirical data, showing a correlation between the strength of inhibition and the frequency of intrinsic gamma oscillations. More importantly, we provide a mechanistic explanation for this correlation, suggesting that it reflects the effects of E-I homeostasis maintaining stable firing rates in cortical areas that receive varying levels of external input or have differences in microcircuitry [7].

The study of the effects of inhibitory synaptic timescales in cortical oscillatory dynamics is not new. Previous studies demonstrate that gamma oscillations can emerge in E-I loops if inhibition has a slower decay than excitation [77], favoring oscillations in the low gamma range [78]. Furthermore, longer inhibitory time constants also favor the emergence of complex oscillatory dynamics, with noise-driven switching between regimes of intermittent and sustained rhythms [79]. However, although it has been suggested that the interaction between synaptic time constants and E-I balance is pivotal in determining cortical rhythms [78], we are the first, to our knowledge, to explore this interaction in systems where the balance is actively maintained through E-I homeostasis. In such circuits, our results suggest that the timescale of inhibitory synapses can influence the bifurcation between damped and sustained oscillations. More specifically, given two systems with the same implementation of homeostasis, target firing rate ($\rho$), and external input ($I^{ext}$), the one with slower inhibition is more likely to engage in sustained oscillations. This effect may be relevant for cortical dynamics, since cortical networks are known to generate slower (alpha/beta) oscillations in deep layers, while faster (gamma) rhythms are observed in superficial layers [67–70]. Therefore, our results on the time constant of inhibition should be interpreted in light of these laminar-specific rhythmic activity patterns. First, under the assumption that deep and superficial layers of the same local circuit have similar target firing rates and input levels, our results would suggest that deeper layers are more likely to engage in persistent oscillations. This effect could be counteracted, for example, by the maintenance of lower target firing rates in deep layers. However, extensive evidence suggests that the opposite is true, with higher spontaneous firing rates in deep versus superficial layers [80–84]. On the other hand, there is also the possibility that deeper layers receive stronger inputs which, when combined with the higher target firing rates, could set them in a similar regime relative to the bifurcation as superficial layers (Fig 8). While models of the canonical

inter-laminar microcircuit account for stronger external inputs to deeper layers [81], detailed anatomical studies suggest instead that the proportion of deep and superficial inputs received by cortical areas depends on their position on the cortical hierarchy [69, 85, 86]. However, in cortical networks, most excitation comes from local circuits [4–8]. In this context, inter-laminar interactions are dominated by excitatory projections from layer 3 to layer 5, while layer 5 neurons strongly project to layer 3 interneurons [69, 81]. It is then reasonable to infer that pyramidal neurons in deeper layers might receive stronger excitatory inputs than their counterparts in superficial layers. Therefore, the stronger inputs and higher mean firing rates in deep layers could compensate for the effect of slower inhibition on the bifurcation of cortical dynamics, posing deep and superficial layers in a similar activity regime relative to the edge-of-bifurcation. This should, however, be studied with the appropriate level of detail. For that reason, we suggest that future studies focus not only on the laminar differences in mean firing rates and levels of input, but also investigate signatures of edge-of-bifurcation dynamics, such as long-range temporal correlations [87, 88], in the slow and fast oscillations of deep and superficial layers. In addition, computational models accounting for the laminar patterns of fast and slow inhibition [69, 70] should be employed to study how inter-laminar interactions, homeostatic plasticity and heterogeneity in external inputs and target firing rates affect the dynamics of local cortical circuits. Finally, it should be pointed out that SST interneurons, which are the main source of slow inhibition to PY neurons [51], show no signatures of participating in E-I homeostasis [12, 21]. Nonetheless, the effect of inhibitory time constants on the bifurcation is evident even in models without homeostasis of inhibition (see $G^E$ or $\mu^E$ panels in Fig 8). Therefore, we argue that it is still worth studying how E-I homeostasis shapes the dynamics of local circuits with fast and slow inhibition.

On another note, it should be stated that this study is not the first to investigate the benefits of employing multiple homeostatic mechanisms on cortical dynamics, although previous computational studies have focused mainly on networks of spiking neurons [26, 89, 90]. Combining the synaptic scaling of excitatory synapses and the plasticity of firing thresholds has been shown to optimize mutual information between neuronal inputs and outputs better than individual homeostatic mechanisms, contributing to the adaptation to perturbations in stimulus statistics [89]. Whether these benefits are present or not at the level of mean population activity remains to be investigated in future studies using our methodological framework. Furthermore, studies in spiking networks demonstrate that each mechanism has a specific contribution to circuit dynamics [90]. More specifically, synaptic scaling is relevant for the recovery of correlations between spontaneous activity. However, our methodology does not allow for the investigation of correlation structures, since population activity is abstracted in a single variable ($r^E$ or $r^I$). Conversely, the plasticity of firing thresholds was found to contribute more strongly to the maintenance of mean firing rates [90]. Here, we argue that this finding is in line with our results suggesting the plasticity of $\mu^E$ can robustly maintain mean firing rates, regardless of input levels, without affecting circuit dynamics (see Fig 4C). That said, we stress that most studies show that the plasticity of intrinsic excitability is rather achieved through the synergistic modulation of both firing thresholds and slopes [23–25]. Therefore, we propose that a mechanism that modulates both $\mu^E$ and $\sigma^E$ is more appropriate, although its functional advantages are not yet clear. In addition, it should be pointed out that further modeling results also suggest that scaling of excitation and inhibition in pyramidal neurons can also regulate firing rates effectively, although the plasticity of intrinsic excitability was not accounted for [26].

In this study, although we derived the mathematical solution for $c^{IE}$ homeostasis, we did not delve deeply into the study of firing-rage homeostasis in inhibitory interneurons. However, research suggests that firing-rate homeostasis in fast-spiking interneurons not only occurs, but is faster than excitatory homeostasis, providing a mechanism for fast regulation of

network dynamics following perturbations [42, 48]. That said, the precise mechanisms of inhibitory firing-rate homeostasis are still unknown. While some results point out that the excitatory synapses in PV neurons are also scaled in response to perturbations in PY firing rates [43, 60], this mechanism may only come into play when activity is higher than the target [43]. Conversely, our analysis suggests that, at least for the WC model, it is not a sufficiently robust mode of homeostasis precisely when perturbations (i.e. higher $I^{ext}$) lead to higher firing rates. This is because homeostasis of $c^{IE}$ acts through the disynaptic modulation of E firing rates by increasing the excitation of the inhibitory population, which provides feedback inhibition. Since inhibitory activity has an upper limit due to the sigmoid non-linearity of the model, this mode of homeostasis breaks down under higher levels of external input (see S5 Appendix and S5 Fig). However, it is possible that this mode of homeostasis, in combination with other forms of plasticity, can contribute to the maintenance of circuit dynamics. For example, the plasticity of inhibitory-to-inhibitory synapses has been shown to contribute to the mainte- nance of network criticality after sensory deprivation in a spiking model of the visual cortex with multiple homeostatic mechanisms [26]. Therefore, given that our model can be adapted to include self-inhibition (S1 Appendix and S1 Fig), we suggest that our approach can be used to study the modulation of both $c^{IE}$ and $c^{II}$ for inhibitory firing-rate homeostasis and its effects on circuit dynamics.

In this work, we have focused on mean-field circuit dynamics, demonstrating the effects of multiple homeostatic mechanisms for the mean activity of populations of thousands of excit- atory and inhibitory neurons. The main reason for this choice is the potential of neural-mass models to be used in the study of the effects of E-I homeostasis in the large-scale dynamics of cortical networks [36, 39–41]. However, such studies have focused mainly on the plasticity of inhibitory synapses, neglecting the potential impact of further homeostatic mechanisms. For that reason, we developed the current framework to be implemented in large-scale models of the human cortex, investigating the specific contributions of different homeostatic mecha- nisms to large-scale cortical dynamics. That said, it should be investigated if our findings extend to more detailed models accounting for neuronal spiking and more complex synaptic dynamics. In this context, we point out that the Wilson-Cowan model has been demonstrated to capture a wide range of aspects of cortical dynamics [45], suggesting that it is an appropriate framework for our study. However, we also point out that homeostatic plasticity has been simi- larly implemented in more detailed spiking and firing-rate models of the cortical neurons [14, 26, 89–91], showing that it can not only maintain the mean firing rate of neurons but also have effects on circuit dynamics, such as the distance-to-bifurcation [26], in line with our results. Therefore, we believe that our conclusions might still apply to more detailed models of cortical networks. Nonetheless, future investigation is warranted, focusing on mechanisms of homeo- stasis that have not yet been implemented in spiking models (e.g. the co-modulation of $\mu^E$ and $\sigma^E$) and in performing a direct comparison between the effects of E-I homeostasis in models that reproduce cortical dynamics at different scales, from the spiking activity of single neurons to whole-brain dynamics.

A further limitation of our approach is that we focus on the slow regulation of E-I balance and circuit dynamics through distinct mechanisms of E-I homeostasis [17, 18]. However, it is known that E-I balance can be modulated in faster timescales. For example, during the forma- tion of neural ensembles, fast E-I balance modulation can be recruited to regulate the speed- accuracy trade-off [16]. Similarly, recent results show that neuromodulatory signals can be used to adjust E-I balance in cortical networks and regulate their sensitivity to noise during decision-making tasks [92] or to control the gating of sensory inputs during learning [93]. Nonetheless, we argue that the slower homeostatic regulation of dynamics towards the edge of bifurcation is essential to support the fast modulation of E-I balance, by ensuring that cortical

networks are poised in a regime where they can be quickly brought to either a state of high or low excitability depending on task demand. For this reason, we suggest that, while homeostasis does not interfere with faster mechanisms of E-I balance modulation, due to the separability of their timescales [17], it might still be essential to ensure that such mechanisms can be quickly recruited to shape circuit dynamics in behaviorally relevant timescales. On another note, it should also be pointed out that homeostatic plasticity is known to interact with other learning mechanisms, such as Hebbian plasticity, contributing to the stabilization of both firing-rates and memory engrams [14, 91]. Nonetheless, other studies highlight the relevance of homeostatic plasticity, suggesting that Hebbian-like learning can emerge in networks with only structural mechanisms of firing-rate homeostasis [94, 95], and that homeostatic plasticity explains the organization of connectivity in motor areas better than Hebbian mechanisms [96].

To conclude, the framework developed and explored in this work benefits the study of the role of E-I balance, and its different homeostatic mechanisms, regarding the emergent circuit dynamics of the cortex. In addition, our extensive study of bifurcation dynamics allows for a deeper understanding of how nodes react in the network context, with implications for network stability and the emergence of collective patterns. More importantly, our results show that the regulation of firing rates in cortical PY neurons can also serve to control edge-of-bifurcation dynamics in cortical circuits. Importantly, the bifurcation between damped and sustained oscillations is shaped differently by distinct modes of homeostasis, offering a plausible explanation for the heterogeneity in distance to the bifurcation found across cortical areas, with likely implications for the behavior of large-scale networks. In addition, we show that E-I homeostasis, particularly when involving inhibitory synapses, has an impact on the frequency of gamma oscillations, which can explain the variation of gamma frequencies with the density of inhibitory receptors across cortical areas. Finally, our results evidence the robustness of relying on multiple modes of homeostasis, which can maintain stable firing rates with minimal perturbations to node dynamics, ensuring that gamma oscillations remain within a physiological range. All in all, we assert the pivotal role of E-I balance in cortical dynamics, posing E-I homeostasis as a robust mechanism for the control of not only firing rates but also edge-of-bifurcation dynamics in cortical networks.

## Supporting information

**S1 Fig. Circuit dynamics in models with different combinations of $c^{II}$ and $\alpha$.** Circuit dynamics of models under homeostasis of $G^E$, $c^{EI}$, $\mu^E$, and $\sigma^E$ with different combinations of external input $I^{ext}$ and fixed point $r^E$. Colors represent the result of linear stability analysis for each combination of parameters, as described by the legend in the top-right. In each sub-plot, we present the results of this analysis for models with a different combination of $c^{II}$, the strength of self-inhibition, and $\alpha$, representing the relative strength of the external input to the inhibitory population, compared to $I^{ext}$ (input to the excitatory neural mass).
(TIF)

**S2 Fig. Activation functions $F(x)$ with different combinations of $\mu^E$ and $\sigma^E = K\mu^E$.** Note that the value of $F(0)$ is always the same as long as $\sigma^E = K\mu^E$.
(TIF)

**S3 Fig. Behavior of the Wilson-Cowan model under homeostasis of $c^{IE}$.** (Left) Homeostatic value of $c^{IE}$ and (Right) corresponding system behavior for different combinations of $r^E_{fixed}$ and $I^{ext}$.
(TIF)

**S4 Fig. Evolution of model parameters under different modes of homeostasis.** Solid lines represent the model parameters at any given point in time. Conversely, dashed lines represent the predicted values. For all simulations, we used $I^{ext} = 2$ and $\rho = 0.1$, and the equations were integrated using the Euler method with a time step of 0.2 ms.
(TIF)

**S5 Fig. $\frac{dc^{EI}}{dt}$ as a function of $r^E$.** We present two cases where $\frac{dc^{EI}}{dt} = r^I(r^E - \rho)$ (solid blue line) or $\frac{dc^{EI}}{dt} = r^E - \rho$ (dashed black line). For illustration purposes, we consider that $\tau_{homeo} = 1s$ and $\rho = 0.1$.
(TIF)

**S6 Fig. Simulated $r^E$ and respective frequency of oscillation under $c^{EI}$ homeostasis for different combinations of $I^{ext}$ and fixed point $r^E$.** The black dashed line represents the analytical prediction for the frequency of oscillation around the fixed point.
(TIF)

**S7 Fig. Model parameters and dynamics for multiple values of $\rho$.** a) Homeostatic value of $G^E$ (Left) and model dynamics (Right) in the model with homeostasis of $G^E$. b) Homeostatic value of $c^{EI}$ (Left) and model dynamics (Right) in the model with homeostasis of $c^{EI}$. c) Homeostatic value of $\mu^E$ (Left) and model dynamics (Right) in the model with homeostasis of $\mu^E$. d) Homeostatic value of $\mu^E$ (Left) and model dynamics (Right) in the model with homeostasis of $\mu^E$ and $\sigma^E$. For all values of $\mu^E$, $\sigma^E = K\mu^E$ e) Homeostatic value of $G^E$ (Top Left), $c^{EI}$ (Top Right) and model dynamics (Bottom) in the model with homeostasis of $G^E$ and $c^{EI}$. f) Homeostatic value of $G^E$ (Top Left), $c^{EI}$ (Top Right), $\mu^E$ (Bottom Left) and model dynamics (Bottom Right) in the model with homeostasis of $G^E$, $c^{EI}$ and $\mu^E$. g) Homeostatic value of $G^E$ (Top Left), $c^{EI}$ (Top Right), $\mu^E$ (Bottom Left) and model dynamics (Bottom Right) in the model with homeostasis of $G^E$, $c^{EI}$, $\mu^E$ and $\sigma^E$. For all values, $\sigma^E = K\mu^E$.
(TIF)

**S8 Fig. Ranges of variation of model parameters under different modes of homeostasis.** a) Value of $G^E$ compared to the default for different modes of homeostasis. Plots show the log difference between $G^E$ and default $G_0^E = 1$ for different combinations of $\rho$ and $I^{ext}$ in models with homeostasis of $G^E$ (Top Left), $G^E + c^{EI}$ (Top Right), $G^E + c^{EI} + \mu^E$ (Bottom Left) and $G^E + c^{EI} + \mu^E + \sigma^E$ (Bottom Right) b) Value of $c^{EI}$ compared to the default for different modes of homeostasis. Plots show the log difference between $c^{EI}$ and default $c_0^{EI} = 2.5$ for different combinations of $\rho$ and $I^{ext}$ in models with homeostasis of $G^E$ (Top Left), $G^E + c^{EI}$ (Top Right), $G^E + c^{EI} + \mu^E$ (Bottom Left) and $G^E + c^{EI} + \mu^E + \sigma^E$ (Bottom Right) c) Value of $\mu^E$ compared to the default for different modes of homeostasis. Plots show the log difference between $\mu^E$ and default $\mu_0^E = 1$ for different combinations of $\rho$ and $I^{ext}$ in models with homeostasis of $G^E$ (Top Left), $G^E + c^{EI}$ (Top Right), $G^E + c^{EI} + \mu^E$ (Bottom Left) and $G^E + c^{EI} + \mu^E + \sigma^E$ (Bottom Right).
(TIF)

**S1 Appendix. Effects of E-I homeostasis in models with self-inhibition and non-zero input to the inhibitory population.**
(PDF)

**S2 Appendix. Derivation of nullclines and fixed point expression.**
(PDF)

**S3 Appendix. Homeostasis of excitatory synapses unto inhibitory neurons.**
(PDF)

**S4 Appendix. Computing steady-state parameters in systems with multiple modes of homeostasis.**
(PDF)

**S5 Appendix. Dynamics of $c^{EI}$ homeostasis.**
(PDF)

**S6 Appendix. Derivation of Jacobian matrix for the Wilson-Cowan model.**
(PDF)

**S7 Appendix. Analytical derivation of oscillation frequency around a fixed point.**
(PDF)

## Author Contributions

**Conceptualization:** Francisco Páscoa dos Santos.

**Formal analysis:** Francisco Páscoa dos Santos.

**Funding acquisition:** Paul F. M. J. Verschure.

**Investigation:** Francisco Páscoa dos Santos.

**Methodology:** Francisco Páscoa dos Santos.

**Software:** Francisco Páscoa dos Santos.

**Supervision:** Paul F. M. J. Verschure.

**Visualization:** Francisco Páscoa dos Santos.

**Writing – original draft:** Francisco Páscoa dos Santos.

**Writing – review & editing:** Francisco Páscoa dos Santos, Paul F. M. J. Verschure.

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
