## [Decision Letter · Decision Letter 0]

11 Oct 2024

Dear Dr. Páscoa dos Santos,

Thank you very much for submitting your manuscript "Excitatory-Inhibitory Homeostasis and Bifurcation Control in the Wilson-Cowan Model of Cortical Dynamics" for consideration at PLOS Computational Biology. As with all papers reviewed by the journal, your manuscript was reviewed by members of the editorial board and by several independent reviewers.

Reviewers emphasized the need for improved clarity in equations and figures, as well as a more thorough discussion. Additionally, a clearer rationale for parameter choices and a deeper exploration of modeling assumptions, especially regarding inhibitory self-coupling, were recommended before publication. You can find additional remarks below and in the attached review file.

Based on the reviews, we are likely to accept this manuscript for publication, providing that you modify the manuscript according to the review recommendations.

Sincerely,

Lorenzo Fontolan, Ph.D.

Guest Editor

PLOS Computational Biology

Andrea E. Martin

Section Editor

PLOS Computational Biology

Reviewer's Responses to Questions

Comments to the Authors:

Please note here if the review is uploaded as an attachment.

Reviewer #1: Recommendation: Revision

Comments to authors:

This study scrutinizes the effects of various modes of E-I homeostasis on cortical dynamics by using well-established concepts of dynamical systems theory. The study design and findings are consistent. The presented results support the claims. The manuscript is well written. There are a few questions and suggestions that the authors are encouraged to take into account:

Majors:

- Page 3, line 83, and Eq. (1): The term G^E in Eq. (1) was introduced to o control the gain of all excitatory inputs. However, in table 1, it was assigned a value equal to 1, but later in line 155, its value was changed and it was linked to the stabilization of excitatory firing rate corresponding to a fixed point under the external current. Was is the implication of G^E = 1?

- Page 5, Fig. 1b: It would be nice to extract the values of r^E in Fig. 1b and indicate them either in the figure or in the caption of the figure.

- This study focuses primarily on how cortical dynamics are shaped by the diverse mechanisms of E-I homeostasis. But, it would be worthy to also discuss the relation of homeostatic plasticity (that stabilizes network activity) to Hebbian plasticity (structural as well functional) that shapes network connectivity (see, for example, Gallinaro & Rotter, 2018, Scientific Reports). It would be crucial to acknowledge that various plasticity mechanisms in different forms work in concert to shape cortical dynamics.

- Page 25, line 634, Effect of time scale of the inhibitory synapses on the emergence of oscillatory dynamics have been focus of several papers that could be refered. See e. g., Khanjanianpak et al, iScience 2024; Roohi and Valizadeh, Neural computation 2023.

Minors:

- Could the authors please re-adjust the figures in the manuscript to be positioned either at the top or bottom of the page?

- Caption of Fig. 2: A period is missing after “… in the Wilson-Cowan Model”.

- Page 7, lines 157-158: “Another common form of E-I homeostasis is the synaptic scaling of fast-spiking 157 inhibitory synapses onto pyramidal neurons [12, 17, 21] (Fig 2).”. Please clearly indicated which panel in Fig. 2.

- Page 8, line 196: “Here, we present a heuristic process or method or something like that …” instead of “Here, we present a heuristic …”.

- Caption of Fig. 3: There is an extra period just after I^ext = 2.

Reviewer #2: The review is uploaded as an attachment.

Reviewer #3: The authors investigate the homeostatic control of firing rates in a model network by a combination of analytical and numerical methods.

In the present study there are 2 novelties with respect to the existing studies: (i) the authors consider the operation of multiple, experimentally-demonstrated, mechanisms of homeostasis (i.e., synaptic scaling of E-to-E and I-to-E connections, and plasticity of intrinsic excitability) both in isolation and in combination; (ii) the authors investigate the 'side effects' of the homeostatic control of the firing rates, that is, how the distance from the bifurcation (between a stable fixed-point and a limit cycle) and the frequency of the oscillations (damped and sustained) are impacted by the different homeostatic mechanisms.

The main result is a proof-of-concept that the distance to criticality (here, distance from the fixed-point-to-limit-cycle bifurcation) can be, indeed, regulated by properly (i.e., by the right combination of homeostatic mechanisms) controlling the firing rates. This is obviously relevant for theories of brain criticality (in a broad sense). The results presented also suggest some of the potential mechanisms by which regional specialization (at the large-scale level) could be obtained starting from 'homogeneous' building blocks (i.e., the local networks).

The study is carefully carried-out, the results are novel and certainly interesting, the manuscript is well written. I have only a few, rather minor, comments.

There are several simplifications/assumptions in the modeling. That is perfectly fine, modeling is essentially about simplifying a problem. However, the potential impact of the simplifications on the results should be clearly discussed.

For instance, the authors point out that in the original model (lines 104-112) the inhibitory population is self-coupled (i.e., inhibitory neurons inhibit themselves besides inhibiting the excitatory neurons). To be sure, inhibitory self-coupling at the population level is also demonstrated experimentally. In the manuscript, they set this coupling to zero, because otherwise the model does not display the bifurcation. However, one could be tempted to argue that, then, perhaps this is not the right model, given that to obtain the behavior of interest I have to neglect well-established experimental information. Similarly, the inhibitory population does not receive external inputs in the model, while it certainly does in the cortex. More generally, to which extent the results depend on the choice of the default parameters (Table 1)? Or alternatively, what is the rationale for this specific choice? The authors only provide a rationale for the choice of the time constants. For instance, inhibitory neurons fire at a much higher rate than excitatory neurons in the cortex.

Eqs. (6), (7) and (10) only make sense if G^E, c^{EI} and \\mu_E are non-negative. Is this obviously always the case? If so, then it should be explained why this is so, and otherwise it should be explained how the non-negativity of the corresponding quantities is ensured, or where is ensured. For instance, it seems to me that G^E in Eq. (6) becomes negative when r^E goes to zero.

Distinguishing dashed from solid lines in, e.g., Fig.4 was extremely complicated for me.

Have the authors made all data and (if applicable) computational code underlying the findings in their manuscript fully available?

Reviewer #1: Yes

Reviewer #2: Yes

Reviewer #3: Yes

PLOS authors have the option to publish the peer review history of their article (what does this mean?). If published, this will include your full peer review and any attached files.

Do you want your identity to be public for this peer review? For information about this choice, including consent withdrawal, please see our Privacy Policy.

Reviewer #1: Yes: Alireza Valizadeh

Reviewer #2: Yes: Guido Gigante

Reviewer #3: No

Figure Files:

Data Requirements:

Reproducibility:

References:

If you need to cite a retracted article, indicate the article’s retracted status in the References list and also include a citation and full reference for the retraction notice.

---

## [Decision Letter · Decision Letter 1]

16 Dec 2024

Dear Mr. Páscoa dos Santos,

We are pleased to inform you that your manuscript 'Excitatory-Inhibitory Homeostasis and Bifurcation Control in the Wilson-Cowan Model of Cortical Dynamics' has been provisionally accepted for publication in PLOS Computational Biology.

Best regards,

Lorenzo Fontolan, Ph.D.

Guest Editor

PLOS Computational Biology

Andrea E. Martin

Section Editor

PLOS Computational Biology

Reviewer's Responses to Questions

**Comments to the Authors:**

Reviewer #1: The authors have adequately addressed my points. I can recomment publication of the manuscript.

Reviewer #2: The authors have meticulously addressed the issues raised in my review, resulting in substantial revisions to their manuscript. These revisions have demonstrably enhanced the clarity and rigor of the paper.

Reviewer #3: The authors have satisfactorily addressed all my, rather minor, comments.

**Have the authors made all data and (if applicable) computational code underlying the findings in their manuscript fully available?**

Reviewer #1: None

Reviewer #2: Yes

Reviewer #3: None

PLOS authors have the option to publish the peer review history of their article (what does this mean?). If published, this will include your full peer review and any attached files.

Reviewer #1: **Yes: **Alireza Valizadeh

Reviewer #2: **Yes: **Guido Gigante

Reviewer #3: No

---

## [Editor Report · Acceptance letter]

29 Dec 2024

PCOMPBIOL-D-24-00968R1 

Excitatory-Inhibitory Homeostasis and Bifurcation Control in the Wilson-Cowan Model of Cortical Dynamics

Dear Dr Páscoa dos Santos,

I am pleased to inform you that your manuscript has been formally accepted for publication in PLOS Computational Biology. Your manuscript is now with our production department and you will be notified of the publication date in due course.

With kind regards,

Dorothy Lannert
